# Genome-wide probing of eukaryotic nascent RNA structure elucidates cotranscriptional folding and its antimutagenic effect

Gongwang Yu[1,2,4], Yao Liu[1,2,4], Zizhang Li[2], Shuyun Deng[2], Zhuoxing Wu[2], Xiaoyu Zhang[1,2], Wenbo Chen[2], Junnan Yang[2], Xiaoshu Chen [1,2,3] & Jian-Rong Yang [1,2,3] ✉

The transcriptional intermediates of RNAs fold into secondary structures with multiple regulatory roles, yet the details of such cotranscriptional RNA folding are largely unresolved in eukaryotes. Here, we present eSPET-seq (Structural Probing of Elongating Transcripts in eukaryotes), a method to assess the cotranscriptional RNA folding in *Saccharomyces cerevisiae*. Our study reveals pervasive structural transitions during cotranscriptional folding and overall structural similarities between nascent and mature RNAs. Furthermore, a combined analysis with genome-wide R-loop and mutation rate approximations provides quantitative evidence for the antimutator effect of nascent RNA folding through competitive inhibition of the R-loops, known to facilitate transcription-associated mutagenesis. Taken together, we present an experimental evaluation of cotranscriptional folding in eukaryotes and demonstrate the antimutator effect of nascent RNA folding. These results suggest genome-wide coupling between the processing and transmission of genetic information through RNA folding.

Secondary structure is a significant feature of RNA molecules, as it is a prerequisite of many RNA functions, such as transcription[1], processing[2], translation[3] and degradation[4]. In the cellular environment, the presence of various RNA-binding proteins and helicases can regulate RNA folding beyond thermodynamic equilibrium dictated by the nucleotide sequence of an RNA molecule[5]. It is therefore not unexpected that different structures might be formed at different stages of the life cycle of RNA molecules[6]. For example, RNA starts to fold once it has been transcribed[7] (i.e., cotranscriptional folding). The structure of nascent RNA formed by such cotranscriptional folding has been shown to regulate alternative splicing[8], as well as the speed of RNA synthesis[9].

Among the various potential functions of RNA secondary structures, the regulatory effect on the spontaneous mutation rate of DNA is

of particular interest[10,11]. Specifically, transcription-associated mutagenesis (TAM) is aggravated by the formation of an R-loop, in which nascent RNA forms a stable RNA-DNA hybrid with the template strand of the DNA, leaving the nontemplate strand of DNA single-stranded/exposed and susceptible to mutagenesis. For a given sequence, RNA–RNA duplexes in the stem of the stem-loop formed by the nascent RNA molecule are thermodynamically more stable than RNA-DNA hybrids with the same base pairing[12]; therefore, strong nascent RNA folding near the transcription site (3′ end of the nascent RNA) might competitively inhibit or dissolve the R-loop if the loop is not too large in the nascent RNA stem-loop. The dissolution of the R-loop allows the template DNA strand to anneal back with the nontemplate DNA strand, thereby reducing TAM[10]. Indeed, our previous experiment in *Saccharomyces cerevisiae* showed that synonymous mutations, which

[1]Advanced Medical Technology Center, The First Affiliated Hospital, Zhongshan School of Medicine, Sun Yat-sen University, Guangzhou 510080, China. [2]Department of Genetics and Biomedical Informatics, Zhongshan School of Medicine, Sun Yat-sen University, Guangzhou 510080, China. [3]Key Laboratory of Tropical Disease Control, Ministry of Education, Sun Yat-sen University, Guangzhou 510080, China. [4]These authors contributed equally: Gongwang Yu, Yao Liu. ✉e-mail: yangjianrong@mail.sysu.edu.cn

computationally predicted as strengthening for a nascent RNA structure, dissolved the R-loop and lowered the mutation rate of a reporter gene by >80% (ref. [10]). Nevertheless, in silico prediction of nascent RNA structure was at best moderately accurate[13], thereby leaving the magnitude of the regulatory effect on the mutation rate by the in vivo nascent RNA structure largely unknown, let alone its generality at the genome scale or in other species such as humans[9]. Additionally, does the antimutator effect of nascent RNA folding contributed to the slower evolution and/or the genetic robustness of highly expressed genes? These questions was never quantitatively answered with genome-wide experimental data despite their broad implications in evolutionary biology[11] and cancer genomics[14].

Besides the antimutator effect, many detail aspects of cotranscriptional nascent RNA folding remained largely unexplored. For example, is it always temporally coupled with the progression of transcription? Is there nascent RNA-specific structure that is absent in mature RNA? Which segment of nascent RNA does display thermodynamic- or kinetic-dictated in vivo folding? High-throughput assessment of nascent RNA structure should help answering these questions. In this context, recent technological advancements, especially the combination of biochemical probing and high-throughput sequencing (HTS), have facilitated genome-wide investigations of RNA secondary structures and their biological functions[5,15–21]. Several studies have assessed the structure of nascent RNAs in eukaryotes[6,22], but their lack of information on transcriptional sites hindered the fine-scale resolution of cotranscriptional folding dynamics. To date, the only HTS-based assays assessing the cotranscriptional folding of nascent RNA is structural probing of elongating transcripts (SPET-seq) introduced in bacteria[7]. However, SPET-seq is based on dimethyl sulfate (DMS), which provides no structural information for guanine (G) or uracil (U), thereby limiting the biological interpretation.

To overcome the technical limitations and investigate the TAM-mitigation effect of nascent RNA structures, we adapted the SPET-seq method to eukaryotes and substitute the DMS with NAI-$N_3$, which modifies all four nucleotides instead of two. Our method, eSPET-seq, simultaneously captures both the NAI-$N_3$ modified single-stranded nucleotides and the transcription site, thus allowing genome-wide assessment of nascent RNA structure during transcription. Experimental data from eSPET-seq in *S. cerevisiae* recapitulate known nascent RNA structures, revealing the dynamics of cotranscriptional folding, as well as its (dis-)similarity with mature RNA structure. More importantly, we find an anticorrelation between nascent RNA folding and spontaneous mutation rate of DNA, which directly supports the R-loop-dependent antimutator effect of nascent RNA folding[10]. Further analyses reveals that nascent RNA folding potentially contributes to the slower evolution and higher genetic robustness of highly expressed yeast genes, as well as the gene-specific mutation load in human cancer. Collectively, our results highlight the biological and evolutionary significance of the nascent RNA structure, in particular its potential regulatory effect on the spontaneous mutation rate.

## Results
### Genome-wide in vivo probing of the nascent RNA structure near the transcription site with eSPET-seq
We aimed to capture the nascent RNA structure near the transcription site in yeast. To this end, we adapted the Structural Probing of Elongating Transcripts (SPET-seq) method previously developed in prokaryotes[7] to eukaryotes with several major improvements (Fig. 1a. See "Methods" for detailed experimental procedures). First, we used NAI-$N_3$, a chemical that is capable of modifying all four nucleotides (adenine (A), U, G, and cytosine (C))[23], to probe single-stranded bases in nascent RNA (Fig. 1a). The full coverage for all nucleotides is an apparent advantage over DMS previously used in SPET-seq[7], which can only modify A and C nucleotides. Second, we enriched the nascent

RNA by using the chromatin fraction after cellular fractionation (Supplementary Fig. 1a–c). Third, we ligated the 3′ hydroxyl terminus of the nascent transcript with a 3′ adapter, which was later paired with a primer for reverse transcription (RT). Nascent RNA segments near the transcription site were enriched during this step due to the presence of terminal phosphates in hydrolysis and degradation products. Next, after RT that should stop at the NAI-$N_3$-modified nucleotide and enrichment for biotinylated molecules, the cDNA was extracted and ligated with a 5′ adapter. Finally, the adapter-linked cDNA was PCR-amplified before being subjected to paired-end HTS (Fig. 1a), in which the forward reads represented the unpaired nucleotides tagged by NAI-$N_3$, and the reverse reads represented the transcription site. As a control for the RT stops triggered by factors other than NAI-$N_3$ modification, we also performed parallel experiments without NAI-$N_3$ treatment, in which biotinylated 3′ adapters were used. This experimental procedure (named eSPET-seq, "e" for eukaryote) collectively enabled localization of single-stranded bases (those at NAI-$N_3$-dependent RT stops), which can be further utilized to approximate the secondary structures of nascent RNA. We would like to emphasize that, strictly speaking, the NAI-$N_3$ reactivities captured by eSPET-seq is not a direct measure of the RNA structure, but the flexibility of RNA nucleotides (as compared to NAz), which was previously proposed as a proxy to RNA secondary structure[18,23]. Throughout the current study, we followed this practice but will highlight potential deviations from such assumption when necessary.

To validate the reliability of eSPET-seq, we performed replicated experiments on the *S. cerevisiae* strain BY4741. We found that the number of reads captured for each gene by eSPET-seq was highly repeatable (Pearson's $R = 0.99$, $P < 10^{-300}$. Fig. 1b). Focusing on genes with ≥ 1 average RT stops per nucleotide (Supplementary Fig. 1d, e), we found that both the NAI-$N_3$-tagged RT stop site (Fig. 1c) and the transcription site (Fig. 1d) were highly consistent between replicated experiments. Indeed, the within-gene site wise Pearson's correlation between replicates was at least 0.40 and 0.65 for the NAI-$N_3$-tagged site and transcription site (Fig. 1c, d), respectively. Additionally, the transcription site captured by eSPET-seq was distributed similar to previous observations made with native elongating transcript sequencing (NET-seq)[24] (Supplementary Fig. 2a) and was comparable between exonic and intronic regions (Supplementary Fig. 2b). We found that approximately 52.6%, 40.8% and 6.6% of eSPET-seq reads were from protein-coding genes, ribosomal RNA (rRNA) genes and other noncoding RNA (ncRNA) genes, respectively, which were quite similar to those previously reported by SPET-seq[7]. These observations clearly demonstrated that eSPET-seq was highly repeatable and effective in capturing nascent RNA segments near the transcription site.

To further confirm the capability of eSPET-seq, we compared the eSPET-seq data with known RNA secondary structures. Using the glutamate tRNA as an example, we used all eSPET-seq read pairs corresponding to the same transcription site, which therefore corresponded to the same transcriptional intermediate, to estimate the distribution of RT stops within the transcriptional intermediate of the tRNA (see "Methods"). Note that the 18 nucleotides at the 3′ end were not considered because they were covered by the transcriptional elongation complex[25] (Supplementary Fig. 3). The eSPET-seq reads clearly indicated the single-stranded region in the known tRNA secondary structure, which was formed cotranscriptionally (Fig. 1e, f).

### Dynamics of cotranscriptional folding revealed by eSPET-seq
Previous analyses[7] based on SPET-seq revealed two distinct modes of cotranscriptional folding in bacteria. On the one hand, 71% of local RNA secondary structures formed as they get transcribed. On the other hand, for the remaining 29% of genomic regions, 5′ halves of long-range helices form transient non-native structures until their 3′ counterparts have been transcribed. Assuming instant folding as the null

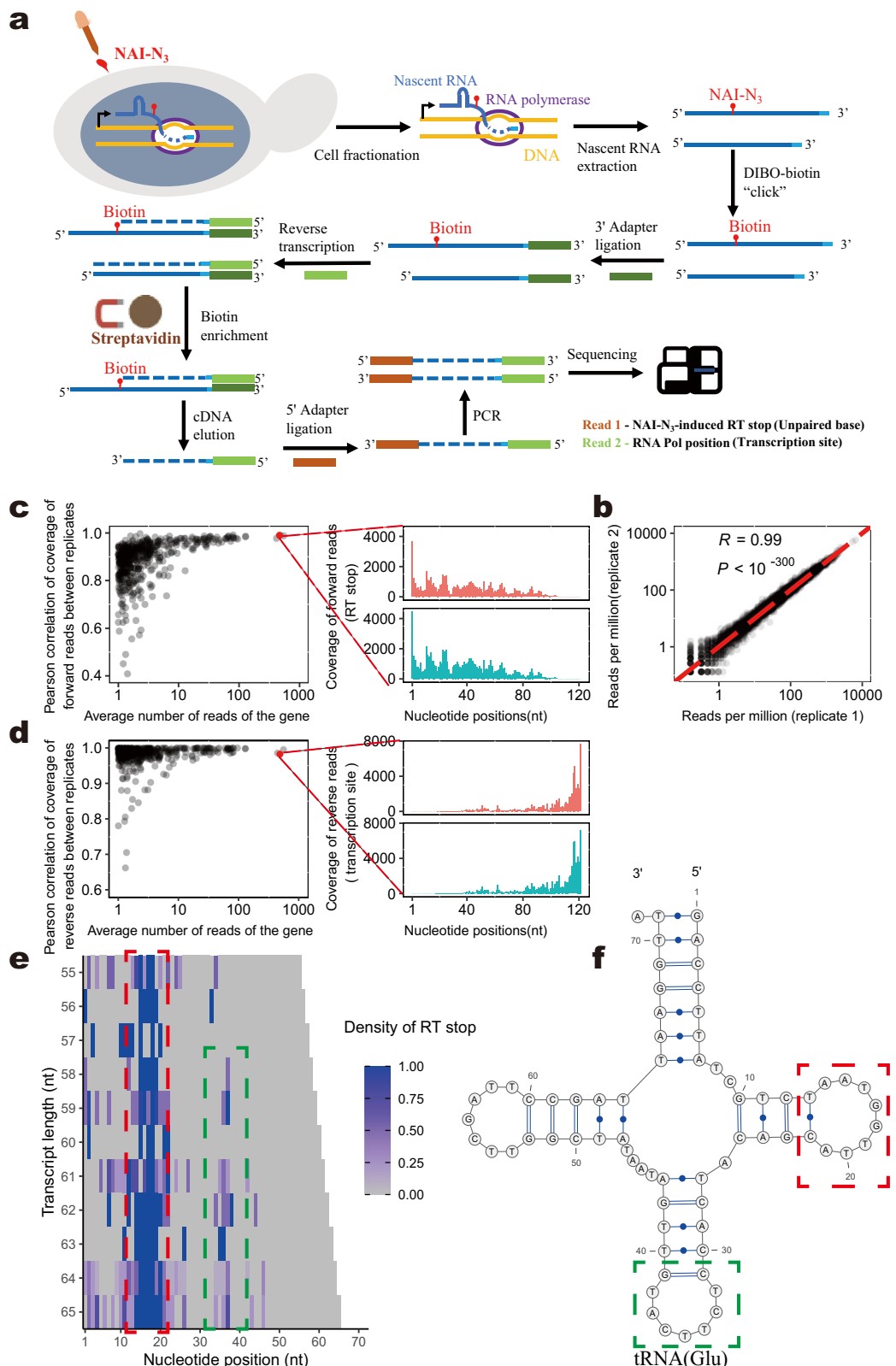

model, we tried to identify the structural transitions by analyzing the changes in the NAI-N$_3$ reactivity as transcription proceeded.

The 5S rRNA was used as an example (Fig. 2a). Similar to the tRNA above, we performed a longitudinal comparison of the RT stop density for the same nucleotide among transcriptional intermediates of different lengths. For example, nucleotides 30–45 had a clear structural transition when the transcript reached a length of ~80 nt, which was

consistent with sequential folding that is made possible only after the downstream half of a stem-loop structure in 5S rRNA is transcribed (Fig. 2b, green box, structural transition indicated by the yellow arrow). Note here that before the structural transition (Fig. 2a, tiles with $y < 80$), the densities of RT stop at nucleotides 30 to 45 were relatively low, a pattern compatible with the existence of an R-loop on 5S rRNA[26], in which the nascent RNA is paired with the template DNA. The

**Fig. 1 | Overview of eSPET-seq procedure and accuracy. a** Experimental procedure of eSPET-seq. **b** The number of reads mapped to each gene was compared between two biological replicates. Pearson's correlation is indicated. The dashed red line represents $x = y$. **c** The within-gene Pearson's correlations ($y$-axis) of the coverage (only the 5′ nucleotide was counted) of the forward reads (representing the RT stops) were calculated between two biological replicates for each gene. The correlation appeared stronger for genes with a higher average number of mapped reads ($x$-axis). The detailed coverage profile is shown for one sample gene (red dot) on the right. **d** Similar to (**c**), except that the correlation calculated for the coverage

(only the 5′ nucleotide was counted) of the reverse reads (representing the transcription site) is shown. **e** Cotranscriptional folding revealed by eSPET-seq. Each row is one transcriptional intermediate of a certain length ($y$-axis). Each column is a specific nucleotide within the gene. The color of each tile represents the normalized (by 90% winsorization) density of the RT stop, which is indicated by the color scale bar. **f** The known structure of the yeast glutamate tRNA. The structures outlined by the red and green boxes correspond to the RT density signals outlined by the red and green boxes, respectively, in (**e**). Source data are provided as a Source Data file.

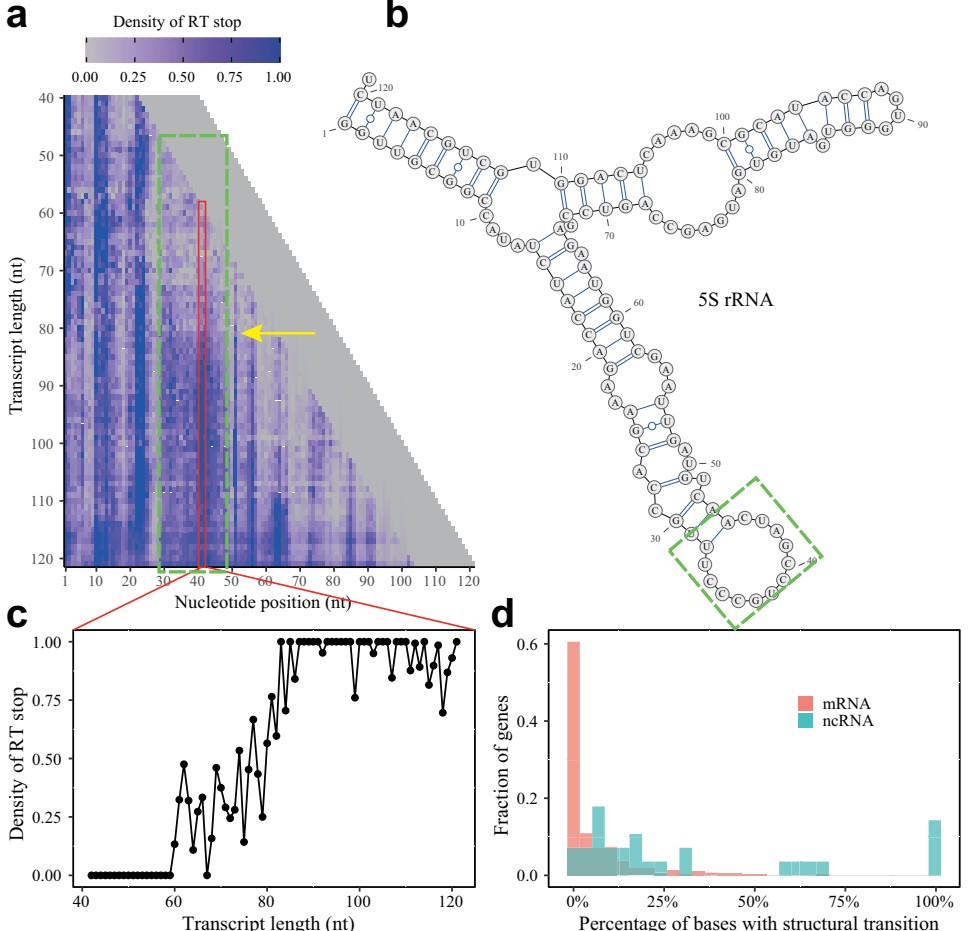

**Fig. 2 | Cotranscriptional folding captured by eSPET-seq. a** The density of the RT stop, as indicated by the color scale bar, is shown for different nucleotides in different transcriptional intermediates of the 5S rRNA. **b** The known secondary structure of the 120 nucleotides of the 5S rRNA. The green box in both (**a**) and (**b**) mark the same region (nucleotides 30–50) with a significant structural transition

indicated by the yellow arrow. **c** The density of the RT stop at nucleotide 42 when 5S rRNA was transcribed from a length of 40–121 nt, which displayed significant structural transition as transcription proceeded. **d** Histogram of the percentage of bases with significant cotranscriptional structural transition for mRNA (red) and ncRNA (green). Source data are provided as a Source Data file.

existence of this R-loop for transcript intermediate <80 nt and its resolution after transcription proceeds to >80 nt in length was further confirmed by DNA:RNA immunoprecipitation(DRIP) followed by reverse transcription quantitative PCR (Supplementary Fig. 4. See "Methods"). For some other regions such as nucleotides 1–25, a structure was formed immediately after it was transcribed, which persisted as transcription proceeded up to the length of the full 121 nt of 5S rRNA (Fig. 2a, tiles with $x$ ranges from 1 to 25). Note that the stem structure between nucleotides 1–9 and 112–120 as observed in the mature 5S rRNA (Fig. 2b) has not formed cotranscriptionally because nucleotide 110-120 remained protected by RNA polymerase II (Supplementary Fig. 3).

To perform a more thorough analysis of structural transitions across the whole yeast transcriptome, we extracted the longitudinal profile of the RT stop density for each nucleotide with the necessary information (example of 5S rRNA is shown in Fig. 2c). We used a sliding window strategy (See "Methods") to locate significant structural transitions as transcription proceeds. Intriguingly, we found that structural transitions were pervasive, as it was observed for an average of 5.4% and 31.7% of nucleotides with necessary information (see "Methods") within mRNA and ncRNA, respectively (Fig. 2d). Although these numbers are probably conservative because further transitions might have been missed by eSPET-seq if they happened too far away from the transcription site, they nevertheless suggested that

ncRNAs tends to have more complex secondary structures that form cotranscriptionally.

## The overall similarity and occasional difference between nascent and mature RNA structures are respectively explainable by cis- and trans-regulation

Other than cotranscriptional folding, another important question about nascent RNA structures was how and why they are similar/different from that of mature RNAs of the same gene. To assess the overall structural similarity between nascent and mature RNA, we performed regular icSHAPE experiments for the in vivo structures of mature RNA. For each of the icSHAPE or eSPET-seq dataset, we extracted all genes with ≥1 average RT stops per nucleotide, and pooled the reads from each gene to estimate its overall structure in the respective states. We found that majority of these genes displayed significant structural similarity between nascent and mature RNAs (Fig. 3a). Here structural similarity was estimated by the Pearson's correlation coefficient between the single-stranded scores obtained for nascent and mature RNAs (Two genes in Fig. 3b, c shown as examples). This observation is compatible with the notion that RNA structures are largely cis-regulated, that is, thermodynamically favored by the RNA sequence.

Furthermore, we hypothesized that trans-regulatory elements, such as RNA-binding proteins, should be responsible for the dissimilarities between nascent and mature RNAs, because the two types of RNA molecules are folded in drastically different cellular contexts (nucleus versus cytoplasm). To test our hypothesis, we conducted additional icSHAPE experiments for the in vitro structures of mature RNA (see "Methods"). We reasoned that, as trans-regulatory factors were depleted during in vitro folding, genes whose NAI-$N_3$ reactivities were similar in vivo and in vitro should have mostly cis-regulated structures, whereas the other genes should have mostly trans-regulated structures. Therefore, our hypothesis predicted that, mature RNA with highly similar in vivo and in vitro structures (i.e. mostly cis-regulated) should also have similar nascent and mature RNA structures folded in vivo, such as observed for *YKLO56C* (Fig. 3b). And vice versa, mature RNA with low similarity between in vivo and in vitro structures (i.e., mostly trans-regulated) should also have dissimilar nascent and mature structures in vivo, such as observed for *YPL250C* (Fig. 3c). Consistent with this prediction, the structural similarity (Pearson's correlation coefficient of the single-stranded scores) between in vivo and in vitro structures of mature RNA is a decent predictor for the structural similarity between nascent and mature RNA structures folded in vivo (Fig. 3d, Spearman's $\rho = 0.32$, $P < 10^{-8}$).

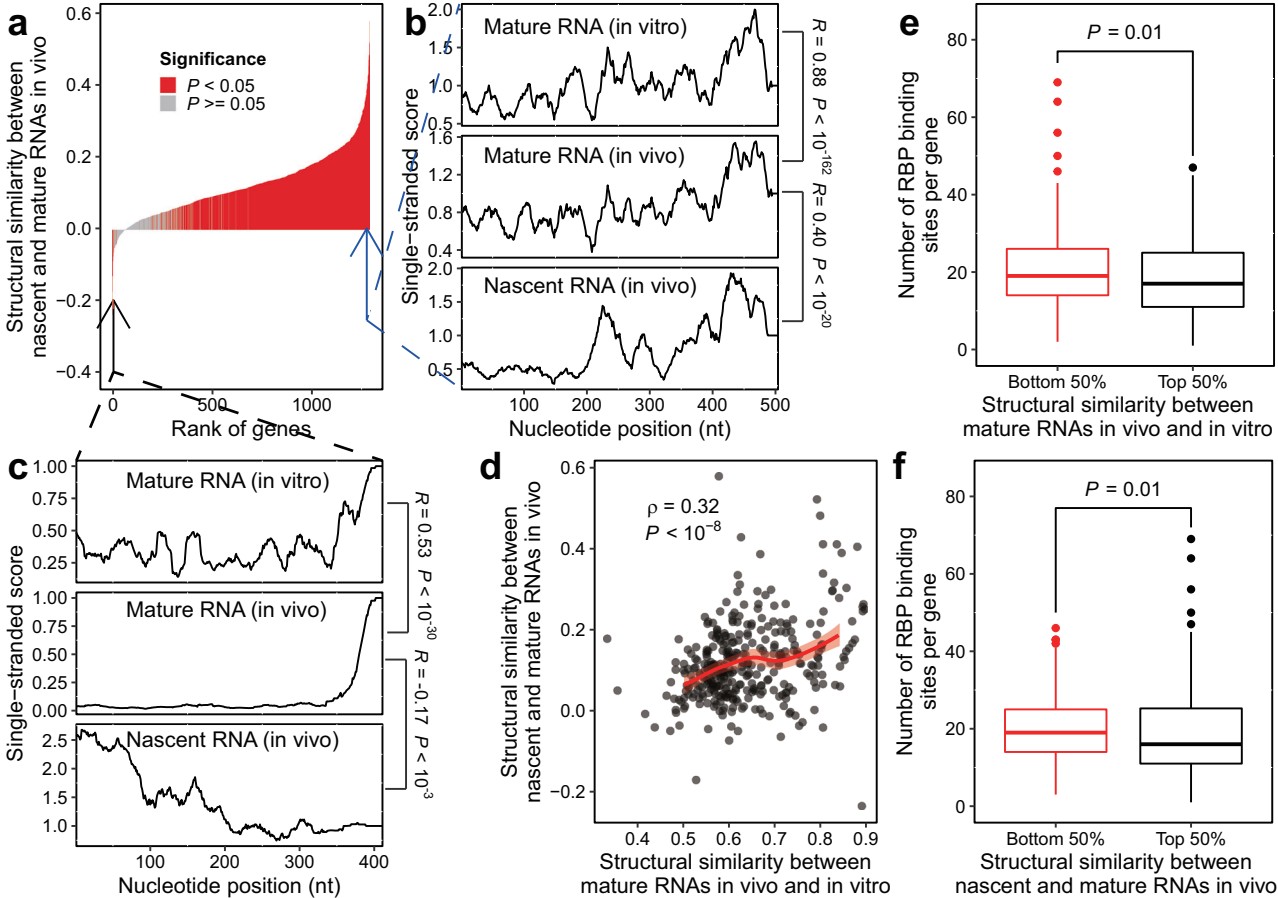

**Fig. 3 | Comparison between nascent and mature RNA structures. a** The structural similarity between nascent and mature RNAs folded in vivo (y axis) was estimated as the within-gene correlation between the corresponding single-stranded scores. Two genes as indicated by the two arrows were chosen as examples, whose single-stranded scores were shown in (**b**) and (**c**). The single-stranded score profiles of nascent RNA folded in vivo, mature RNA folded in vivo and in vitro were shown for *YKLO56C* (**b**) and *YPL250C* (**c**). **d** Structural similarities between mature RNAs folded in vivo and in vitro were compared with that between nascent and mature RNAs folded in vivo. Spearman's rank correlation coefficient is indicated (*n* = 346). **e** Standard boxplot showing that the top 50% of genes with higher structural similarity between mature RNAs folded in vivo and in vitro tended to have less RBP binding sites than the other genes (*n* = 345). **f** Same as (**e**), except that the genes were divided into two equal-sized groups based on the structural similarity between nascent and mature RNAs folded in vivo (*n* = 345). For both (**e**) and (**f**), *P*-values are from the Wilcoxon signed-rank tests. Data are presented as standard box-and-whisker plots, in which the boxes extend from the lower to the upper quartile (interquartile range, or IQR), the thick horizontal line represents the median, whiskers indicate the lowest and highest datum within 1.5*IQR of the lower and upper quartiles, and circles represent outliers outside of the whiskers. Source data are provided as a Source Data file.

To further assess the molecular mechanism underlying the above observation, we downloaded a list of in vivo interactions between protein and mature RNAs experimentally inferred by gPAR-CLIP (global Photoactivatable-Ribonucleoside-enhanced crosslinking and Immunoprecipitation) from a previous report[27]. Using this dataset, we found that the 50% genes with low structural dissimilarities between in vivo and in vitro mature RNA (red-colored in Fig. 3e) tended to have more protein binding sites compared the other 50% genes ($P = 0.01$, Mann–Whitney test). Similarly, the 50% genes with low structure dissimilarities between nascent and mature structures in vivo (red-colored in Fig. 3f) tended to have more protein binding sites compared the other 50% genes ($P = 0.01$, Mann–Whitney test). These results suggested that at least some of the unique nascent RNA structures when compared to the mature RNAs can be explained by the unique cellular contexts in which nascent RNAs folded, including the availabilities of RNA-binding proteins.

## Relationship among mutation rate of a gene, prevalence of nascent RNA structure, and R-loop score

Our previous study[10], on one yeast gene (*CAN1*), has shown that nascent RNA folding alleviates mutagenesis by weakening the R-loop during transcription (Fig. 4a). Nevertheless, genome-wide assessment of the relationship between experimentally measured nascent RNA structure and R-loop or spontaneous mutation rate has not been possible so far. eSPET-seq data offered a unique opportunity to resolve this and better quantify the antimutator effect of nascent RNA folding at the genome scale. To this end, we gauged the spontaneous DNA mutation rate by the population genomic data from 190 *S. cerevisiae* strains[28] (see "Methods"), the propensity to form R-loops by an "R-loop score" derived from yeast S1 nuclease DNA-RNA immunoprecipitation with deep sequencing[29] (see "Methods"), and the prevalence of nascent RNA structure by a single-stranded score estimated from the eSPET-seq data. Here the single-stranded score was first calculated for each nucleotide (see "Methods"), reflecting the relative probability of it being unpaired/single-stranded, and then aggregated for a per-gene metric of prevalence of nascent RNA structure by the negated mean within the gene or by Gini index (a higher Gini index indicates a more structured region[5]) (Supplementary Fig. 5a, b). A single-stranded score can also reveal differences in the prevalence of nascent RNA structure between sub-genic regions, such as exons being more folded than introns (Supplementary Fig. 5c, d). We would like to emphasize here again that the NAI-N$_3$ reactivity captured by eSPET-seq only approximates the prevalence of single-stranded RNA, therefore any metrics derived from the single-stranded scores should be interpreted as the (un-)prevalence of nascent RNA structure, but not thermodynamic stability of the secondary structure of nascent RNA.

Consistent with our hypothesized model[10], the genes with higher prevalence of nascent RNA structure indeed had lower R-loop scores (Fig. 4b, c). Notably, the corresponding correlations (Spearman's $\rho = -0.198$ and $-0.345$, $P < 10^{-13}$ and $10^{-42}$, respectively) were more than an order of magnitude stronger than previous observations made with computationally predicted nascent RNA structure[10] (Spearman's $\rho = -0.014$). Then, after confirming the expected relationship between the spontaneous DNA mutation rate and the R-loop score (Fig. 4d), we also found that the prevalence of nascent RNA structure was anticorrelated with mutation rate (Spearman's $\rho = -0.219$ and $-0.156$, $P < 10^{-19}$ and $10^{-9}$, respectively, Fig. 4e, f), which were also much stronger than the prediction-based observation[10] (Spearman's $\rho = -0.05$). More importantly, the correlation between the prevalence of nascent RNA structure and R-loop score or mutation rate remained mostly significant if the mature RNA structure was controlled (Supplementary Fig. 6. See "Methods"), suggesting that the R-loop dissolution and mutation mitigation effects of nascent RNA folding were not confounded by mature RNA structure.

To further investigate the role of the R-loop in mediating the antimutator effects of the nascent RNA structure, we calculated the partial correlation between the prevalence of nascent RNA structure and mutation rate, controlling the R-loop score. These partial correlations were $\rho = -0.177$ ($P < 10^{-11}$) for the negative single-stranded score and $\rho = -0.086$ ($P < 10^{-3}$) for the Gini index of the single-stranded score, which respectively suggested that approximately 34.7% ($= 1 - (-0.177^2/-0.219^2)$) and 69.6% ($= 1 - (-0.086^2/-0.156^2)$) of the antimutator effects of the nascent RNA folding were mediated via the R-loop (Fig. 4g). Taking into account the considerable amount of experimental noise, we suspected that these percentages were likely underestimated, such that a large fraction of the antimutator effect of nascent folding was mediated via the R-loop.

Genomic analyses by comparison among genes might be biased by confounding factors such as gene expression level, which can be excluded by within-gene analysis (an example shown in Fig. 4h). We therefore calculated, within each gene, three odds ratios (*OR*) to assess the relationship between the three estimates, such that an excess of sites in support of our hypothesized model (Fig. 4a) will result in an observation of $OR > 1$. They included $OR_{S-R}$ for the relationship between *s*ingle-stranded score and *R*-loop score, $OR_{R-M}$ for that between *R*-loop score and *m*utation rate, and $OR_{S-M}$ for that between *s*ingle-stranded score and *m*utation rate (see "Methods"). For each of the three *OR*s, we combined the *OR*s from all genes using Cochran–Mantel–Haenszel chi-squared test (MH test. See "Methods") and found all three combined *OR*s to be significantly greater than 1 (Fig. 4i). In addition, we calculated the average within-gene correlations between the three estimates, and found them as significantly different from their respective null distribution in a way that is consistent with our hypothesis (Fig. 4j). Here the null distribution is assessed by individually shuffling the three estimates within each gene for 1000 times. These results were further supported by the observation that sites mutated in mutation accumulation experiments[30–32] tend to have lower prevalence of nascent RNA structure compared to non-mutated sites in the same gene (Supplementary Fig. 7). Collectively, our analyses revealed the anticorrelation between prevalence of nascent RNA structure and R-loop or DNA spontaneous mutation rate, thereby offered an experiment-based, quantitative evidence for the antimutator effect of nascent RNA folding at genome scale.

To further confirm the antimutator effect of nascent RNA folding via inhibition of R-loop, we conducted manipulative experiments targeting the *CAN1* gene, which encodes an arginine permease that is commonly used to detect spontaneous mutations[10] and displayed a clear signal of R-loop at the first 300 nt of its 5′-end (Fig. 5a). Three different synonymous mutants of *CAN1* with strong, intermediate and weak nascent RNA folding (Fig. 5b. See "Methods") were expressed in yeast cells. We found that the spontaneous mutation rates were indeed lower in *CAN1* mutants with stronger nascent RNA folding (Fig. 5c). To further verify that the reduction of mutation rate by nascent RNA folding is mediated by inhibiting R-loop formation, we examined the prevalence of R-loops at the 5′ end of these three *CAN1* mutants by DRIP-RT-qPCR, and found that the R-loop signal was significantly reduced when nascent RNA folding became stronger (Fig. 5d. See "Methods"). More importantly, when we stably overexpressed *RNA-SEH1*, which hampers R-loop formation by degrading the RNA in an R-loop, the differences of mutation rate (Fig. 5e) or R-loop prevalence (Fig. 5f) among the three versions of *CAN1* were all weakened or disappeared. Collectively, these *CAN1*-based manipulative experiments demonstrated the R-loop-dependent antimutator effect of nascent RNA folding.

## Genes functionally sensitive to mutations have stronger nascent RNA folding

What could be the biological consequence of such antimutator effect of nascent RNA folding? One question, i.e., whether this antimutator

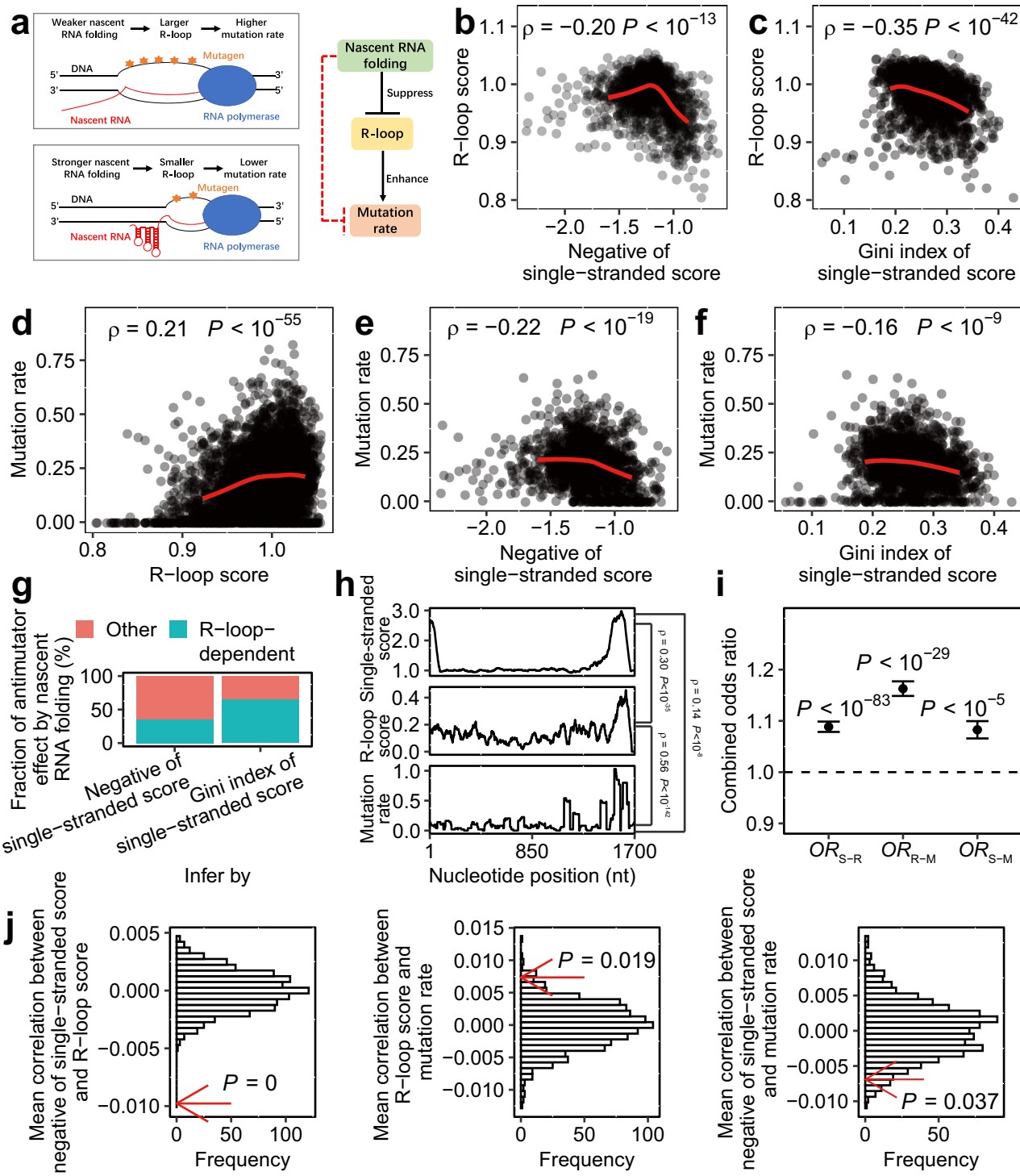

**Fig. 4 | Correspondence among nascent RNA folding, R-loop and mutation rate.**
**a** Schematic diagram showing the hypothesized mechanism underlying the antimutator effect of nascent RNA folding (left) and the resulting relationship between nascent RNA folding, R-loop and mutation rate (right). **b, c** Both estimates of prevalence of nascent RNA structure were compared with the R-loop score of the same gene ($n = 1520$). **d** The R-loop score of a gene was compared with the mutation rate of the gene ($n = 5691$). **e, f** Both estimates of prevalence of nascent RNA structure were compared with the mutation rate of the same gene ($n = 1728$). **g** The R-loop dependent fraction of antimutator effect by nascent RNA folding was estimated by contrasting the correlation and the partial correlation (with R-loop score controlled) between the mutation rate and the prevalence of nascent RNA structure. **h** Example demonstrating the within-gene

correspondence among the single-stranded score, R-loop score and mutation rate of the nucleotide positions. **i** Three odds ratios representing the correspondence among *s*ingle-stranded scores, *R*-loop scores and *m*utation rates ($OR_{S-R}$, $OR_{R-M}$ and $OR_{S-M}$) were calculated for each gene, and then combined and tested for statistical significance by the Cochran–Mantel–Haenszel chi-squared test. Error bars represent s.d. of the combined $OR$ estimated by bootstrapping the genes 1000 times. **j** Three within-gene correlations among negative of single-stranded score, R-loop score and mutation rate of nucleotide positions were averaged (red arrow) and compared with their random expectations (histogram), which were estimated by permuting the negative of single-stranded score, R-loop score and mutation rate within each gene 1000 times. *P* values from permutation tests are indicated. Source data are provided as a Source Data file.

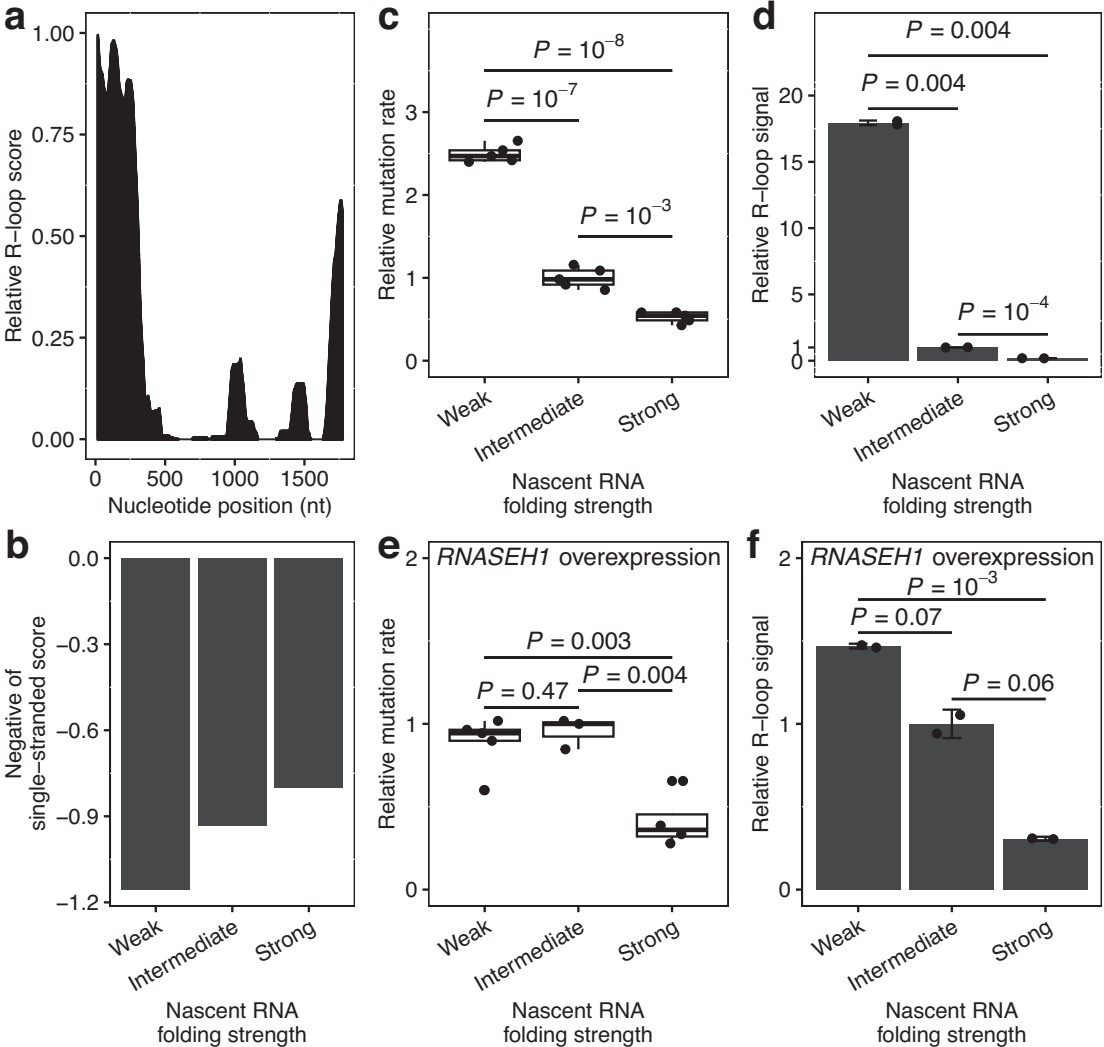

**Fig. 5 | Validating the R-loop-dependent antimutator effect of nascent RNA folding via manipulative experiments. a** Snapshot of R-loop signals from S1-DRIP-seq. The y-axis represents the enrichment of S1-DRIP-seq reads relative to Input chromatin. There is a strong R-loop signal in the 1–300 bp region. **b** Experimentally determined nascent RNA folding strengths of the three selected *CAN1* mutants with weak, intermediate and strong nascent RNA folding. **c** Relative mutation rate of *CAN1* for weak ($n = 5$), intermediate ($n = 5$) and strong ($n = 5$) versions. Data are presented as standard box-and-whisker plots defined as in Fig. 3. **d** Relative R-loop

signal at *CAN1* for weak, intermediate and strong versions. The results are expressed as means ± s.d. of two independent experiments. **e** Relative mutation rate at *CAN1* for weak ($n = 5$), intermediate ($n = 3$) and strong ($n = 4$) versions when *RNASEH1* was overexpressed. Data are presented as standard box-and-whisker plots defined as in Fig. 3. **f** Relative R-loop signal on *CAN1* for weak, intermediate and strong versions when *RNASEH1* was overexpressed. The results are expressed as means ± s.d. of two independent experiments. All *P*-values are based on two-tailed t tests. Source data are provided as a Source Data file.

effect was correlated with gene expression level, was of particular interest for several reasons. First, TAM is by definition more severe for highly expressed genes compared to lowly expressed genes[33,34]. This trend might be exaggerated/weakened by a negative/positive correlation between gene expression and prevalence of nascent RNA structure, thereby affecting the relative evolutionary rate of highly versus lowly expressed genes[35,36]. Second, highly expressed genes were generally more functionally constrained[35] because, relative to lowly expressed genes, a larger fraction of mutations in highly expressed genes were functionally prohibitive[37]. Therefore, more/less prevalent nascent RNA structure in highly expressed genes might contribute to the genetic robustness/fragility. Last but not least, the nascent RNA structure of some genes, due to their intensive transcription, might have a large effect size for TAM-mitigation, such that the nascent RNA structure qualifies as an adaptive gene-specific mutation rate modifier[11,38] (but see below).

To answer the first two questions above, we estimated for yeast protein-coding genes the expression levels, and, as a proxy for

functional constraints, the evolutionary conservation of their sequences (see "Methods"). When these two metrics were compared with prevalence of nascent RNA structure, we consistently found positive correlations (Spearman's $\rho = 0.19$–$0.56$, Fig. 6a–d). More importantly, these correlations remained significant when the prevalence of structure in mature RNA was controlled (Supplementary Fig. 8), suggesting that the observed correlations were not confounded by the more prevalent mRNA structures of highly expressed genes[3,39]. Collectively, the above results suggested that highly expressed genes tended to have more prevalent nascent RNA structures, and therefore might have contributed to the slower evolution and genetic robustness of highly expressed genes.

As for the third question, we need to estimate the effect size of TAM-mitigation due to nascent RNA folding. Within each gene, we calculated the fold reduction in the spontaneous mutation rate between the one 50-bp segment with the highest prevalence of nascent RNA structure, versus the one 50-bp segment with the lowest prevalence of nascent RNA structure (see "Methods"). Intriguingly,

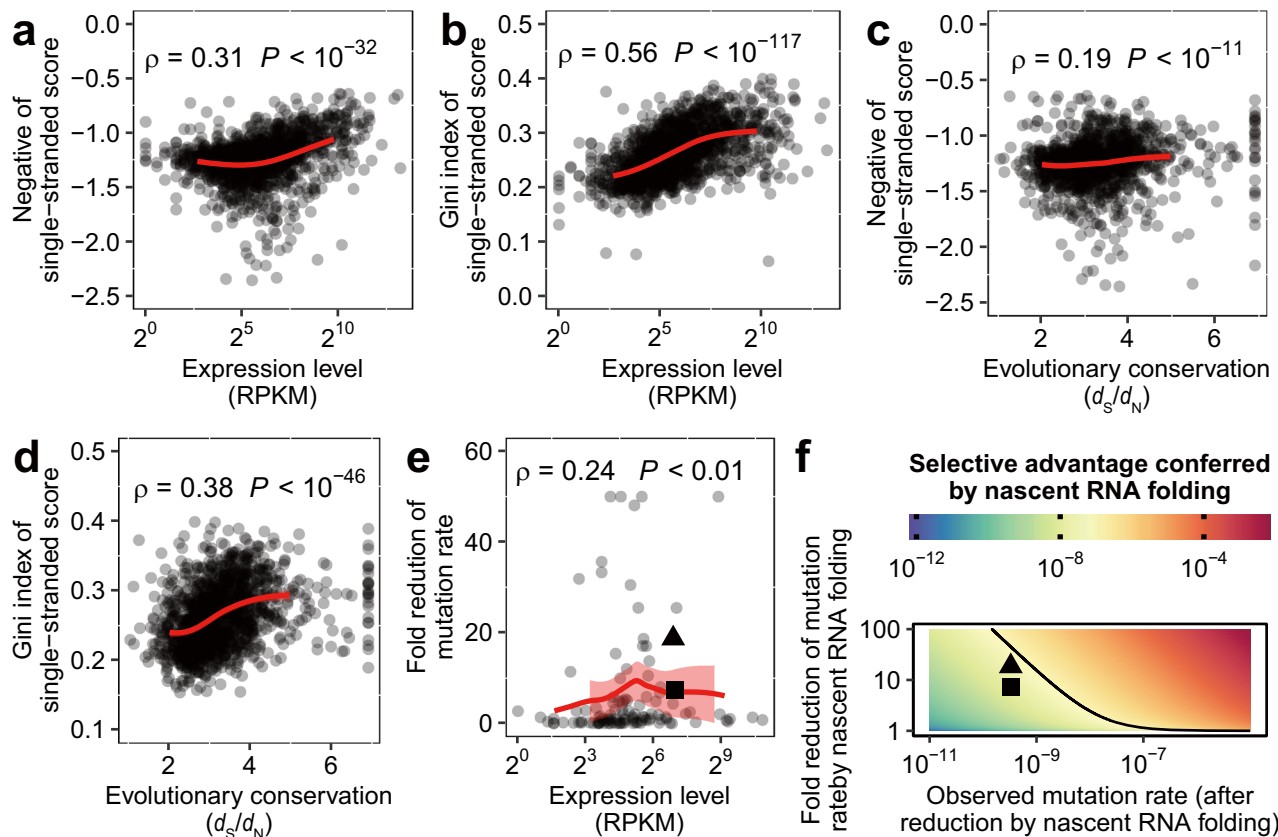

**Fig. 6 | Stronger antimutator effect of nascent RNA folding in highly expressed genes. a–d** The expression level (**a**, **b**) ($n$ = 1404) in unit of Reads Per Kilobase per Million mapped reads, or RPKM, and evolutionary conservation (**c**, **d**) ($n$ = 1311), calculated as the ratio between $d_S$ (the number of synonymous substitutions per synonymous site) and $d_N$ (the number of nonsynonymous substitutions per non-synonymous site. See "Methods") of a gene, were compared to both estimates of prevalence of nascent RNA structure. Spearman's rank correlation coefficient is indicated, and the red lines represent the fitted LOESS regression. The evolutionary conservation was calculated as the log of inversion of $d_N/d_S$ (see "Methods").
**e** Within 113 genes containing at least 5% sites with mutation rate data, the ratio between the mutation rate of a 50-bp segment with the highest single-stranded score and that of a 50-bp segment with the lowest single-stranded score within the same gene was calculated. We used this ratio to approximate the fold reduction in the mutation rate due to the nascent RNA folding for each gene and compared it

with the expression level of the gene. Spearman's rank correlation coefficient is indicated, the red line represents the fitted LOESS regression, and the red shade shows the standard error of the fitted value ($n$ = 113). **f** Phase diagram showing the selective advantage (color scale bar) conferred by the mutation rate reduction due to nascent RNA folding. This selective coefficient was dependent on two critical parameters, namely the mutation rate ($x$-axis) and the fold reduction in the mutation rate by nascent RNA folding ($y$-axis). The solid line indicates the selective coefficient that is equivalent to the inversions of the effective population size of yeast. That is, the parameter space above the lines represents a selective advantage large enough as a subject of natural selection. Two example genes were marked in both (**e**) and (**f**) as a square (*YHL033C*) and as a triangle (*YDR233C*). These two genes show that even for genes showing strong antimutator effect of nascent RNA folding, mutations strengthening the nascent RNA structure might still not be adaptive. Source data are provided as a Source Data file.

we found that fold reduction of mutation rate appeared greater in highly expressed genes (Fig. 6e), suggesting that the TAM-mitigation effect of nascent RNA folding was indeed stronger for genes transcribed more frequently. Nevertheless, the fold reduction in mutation rate per gene (6.4 ± 12) is too small, such that vast majority of the genes displayed <40 fold reduction in mutation rate, and therefore shall only lead to a very small selective advantage (<$10^{-7}$). This selective advantage was smaller than the inversion of effective population size of yeast ($10^7$) (ref. [10]), therefore the increased nascent RNA folding would not be selected as an adaptive trait in yeast (Fig. 6f, see "Methods"). This observation is consistent with previous theoretical work suggesting that selection for mutation rate reduction of individual genes should be generally weak[40]. Altogether, we concluded that antimutator effect of nascent RNA folding is not a direct subject of natural selection, whereas the reason behind its correlation with gene expression remained unresolved. This result also explained why delayed cotranscriptional folding (structural transition during transcription) can be pervasive, as the fitness cost due to delayed cotranscriptional folding and therefore increased TAM were not strong enough. Collectively,

our quantitative analyses of the antimutator effect of nascent RNA folding demonstrated again the potential of eSPET-seq in furthering our biological understanding of nascent RNA structure.

## Discussion

In the current study, we developed eSPET-seq, an experimental assay specifically designed to probe the pairing status of all four RNA nucleotides (A, U, G, and C) near the transcription site, thereby revealing nascent RNA structures. Based on the eSPET-seq data, we found heterogeneous rate of cotranscriptional folding, the (dis-)similarity between nascent and mature RNA structure partially explainable by RNA-binding proteins, and quantitative genomic evidence for the hypothesis that nascent RNA folding mitigates TAM by dissolving the R-loop. eSPET-seq therefore facilitated understandings of nascent RNA folding in eukaryotes with unprecedented resolution, which has broad implications in molecular and evolutionary biology.

From a technological perspective, eSPET-seq had several unique advantages over similar techniques previously developed. Specifically, SPET-seq has been used to probe cotranscriptional folding of nascent

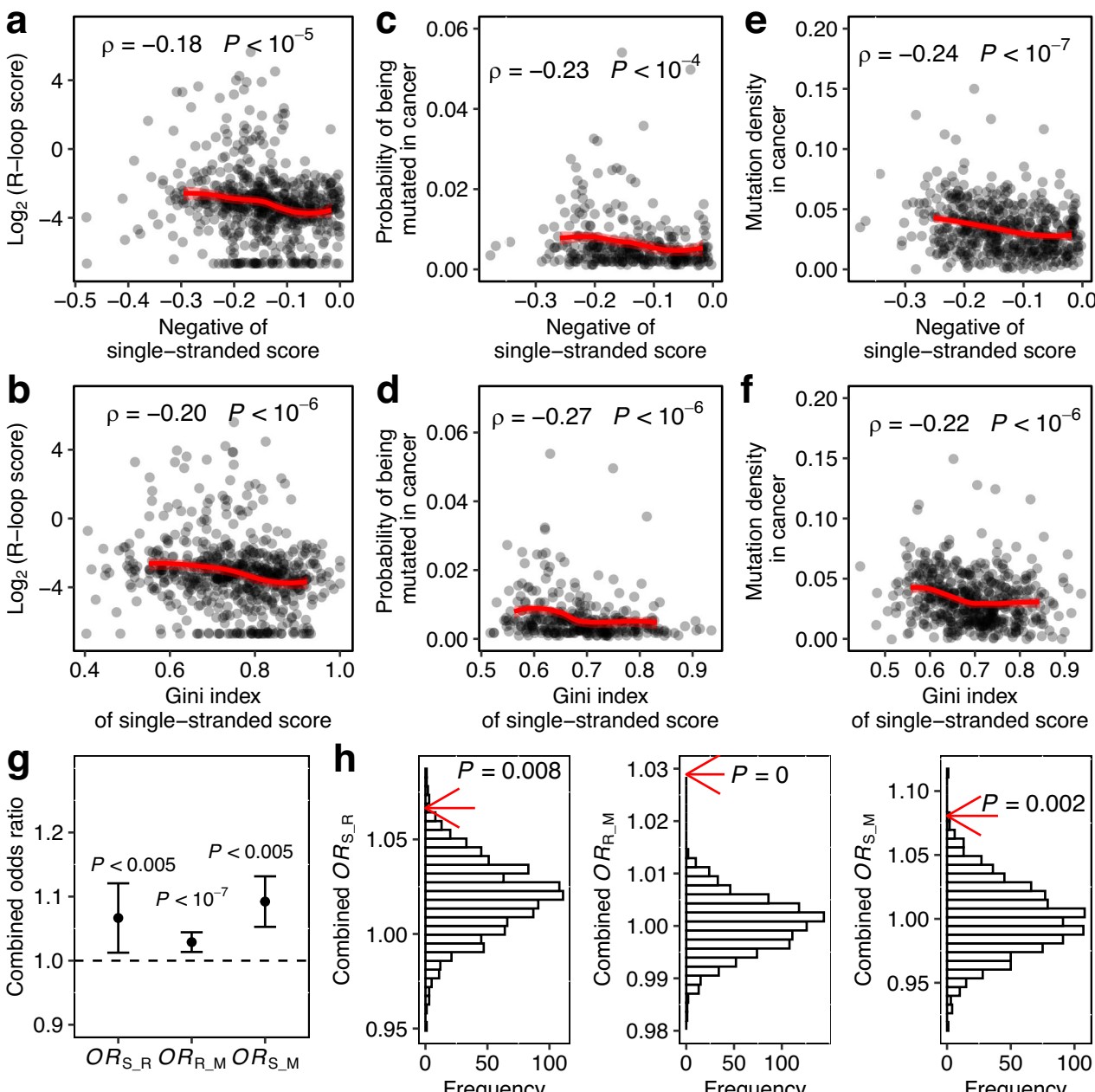

**Fig. 7 | The prevalence of nascent RNA folding of a gene was correlated with its mutation rate in tumors. a–f** Both estimates of prevalence of nascent RNA structure of a gene were compared with its R-loop score (**a**, **b**) ($n = 628$), its probability of being mutated (**c**, **d**) ($n = 315$) and its mutation density (**e**, **f**) ($n = 515$) in cancer. Spearman's rank correlation coefficients are indicated, and the red lines represent the fitted LOESS regression. **g** Three odds ratios representing the correspondence among *single*-stranded scores, *R*-loop scores and *m*utation rates ($OR_{S-R}$, $OR_{R-M}$ and $OR_{S-M}$. See "Methods") were calculated for each gene, and then combined and tested for statistical significance by the Cochran-Mantel-Haenszel chi-squared test. Error bars represent s.d. of the combined $OR$ estimated by bootstrapping the genes 1000 times. **h** The three odds ratios were compared with their random expectations (histogram), which were estimated by permutating the single-stranded score, R-loop score and mutation rate within each gene 1000 times. $P$ values from permutation tests are indicated. Source data are provided as a Source Data file.

RNA in bacteria[7]. However, SPET-seq can only capture the pairing status of A and C nucleotides, but not U and G nucleotides, and therefore has a limited resolution. In addition, icSHAPE has been applied to chromatin-associated nascent RNAs in human HEK293 cells[6]. However, since NAI-N$_3$-tagged sites across the whole nascent transcript was captured without enriching those near the transcription site, the generated data, compared to eSPET-seq data, should be less informative for nascent RNA folding ocurring immediately after transcription, a time point that was presumably critical for R-loop dissolution by nascent RNA folding. In other words, eSPET-seq is, to the best of our knowledge, the first experimental technique assessing the nascent RNA structure for all four RNA nucleotides near the transcription site.

With the help of eSPET-seq, several significant features of nascent RNA folding were revealed. For example, an average of 5.4% of nucleotides within mRNA displayed significant structural transition as transcription progressed. Compared to previous observation of 29% in bacteria revealed by SPET-seq, we speculated that the difference was mainly caused by separation of transcription and translation in eukaryotes, such that structural transitions accommodating

translation are unnecessary until mRNA had finished its transcription and have been exported to cytoplasm in eukaryotes, whereas similar translation-accommodating structural transitions happened soon after transcription and before translation in prokaryotes. Additionally, our comparison between nascent and mature RNA structure in vivo and in vitro suggested that their differences are at least partially explainable by trans-regulatory elements such as RNA-binding proteins. Collectively, data of eSPET-seq has offered a unique experimental approach for the studies of molecular biology of nascent RNA structure, and shall therefore lay the foundation for further studies.

We inferred with eSPET-seq data that the increased TAM in highly expressed genes was weaken by the stronger nascent RNA folding (Fig. 6a, b). This observation could have been confounded by transcription-coupled repair (TCR), as TCR is theoretically more active for highly expressed genes[41] (but see ref. 42). To disentangle the antimutator effect of nascent RNA folding and that of TCR, we exploited the different strand biases of the two mechanisms. Specifically, TCR repairs mutations on the template strand[43] and nascent RNA folding suppresses mutations on the coding strand[10]. We collected spontaneous mutation events found in three mutation accumulation experiments[30–32] and classified all C/G to T/A mutations into either template or coding strand by assuming dominate contribution by hydrolytic deamination of cytosine in C/G to T/A mutations[44] (see "Methods"). Based on these stranded mutations, we found that the prevalence of nascent RNA structure was anticorrelated with the relative C-to-T mutation rate of the coding strand (Supplementary Fig. 9a. Spearman's $\rho = -0.28$, $P < 0.02$) but not that of the template strand (Supplementary Fig. 9b. Spearman's $\rho = -0.02$, $P = 0.83$), thereby suggesting the TCR-independent antimutator effect of nascent RNA folding on the coding strand. In terms of effect size, comparing the top and bottom 20% of genes in terms of prevalence of nascent RNA structure revealed a 25% decrease of C-to-T mutation rate of the coding strand. We further contrast the C-to-T mutation rate of template and coding strand to better isolate the TCR-independent antimutator effect of nascent RNA structure (Supplementary Fig. 9c, d). Specifically, we found that C-to-T mutation rate of template strand (suppressed by TCR) relative to that of the coding strand (suppressed by nascent RNA folding) decreased 10% for highly expressed genes compared to lowly expressed genes (Supplementary Fig. 9c), which revealed the effect of TCR; And the C-to-T mutation rate of coding strand (suppressed by nascent RNA folding) relative to that of the template strand (suppressed by TCR) decrease by 29% for genes with high prevalence of nascent RNA structure compared to genes with low prevalence of nascent RNA structure (Supplementary Fig. 9d), which revealed the antimutator effect of nascent RNA folding. Collectively, these results suggested substantial antimutator effect by nascent RNA folding independent of TCR.

Together with our manipulative experiment for one yeast gene, the genomic correlations among nascent RNA structure, R-loop and mutation rate have collectively supported the antimutator effect of nascent RNA folding, especially for highly expressed genes. In addition to the evolutionary significance mentioned above, regulators for the spontaneous mutation rate might also has broad implications in cancer biology. For example, a temporary increase in mutation rates, i.e., a mutator phenotype, has been proposed as an early step in carcinogenesis[45]. With the mutation mitigation effect of nascent RNA folding confirmed in *S. cerevisiae*, we speculated a similar role for nascent RNA folding in humans. Notably, our speculation was consistent with recent progress in R-loop biology showing that aberrant or excessive R-loop formation can lead to genomic instability, a hallmark of cancer[46], thereby laying the mechanistic foundation for the relevance of the nascent RNA structure. As a further support for our speculation, we analyzed public available relevant data[6] and found that both proxies for prevalence of nascent RNA structure in a gene, i.e. negative of and Gini index of single-stranded score, were negatively correlated with each of R-loop score, probability of being mutated in cancer and mutation density in cancer of the gene (Fig. 7a–f. See "Methods"). More importantly, these patterns found by among-gene comparisons were also consistently supported by within-gene analyses based on combined Odds Ratios using MH tests (Fig. 7g, h. See "Methods"). Collectively, these observations supported the mutation mitigation role of nascent RNA folding in humans, and suggested a potential mechanistic link between the nascent RNA structure and mutational hotspots in cancer.

## Methods

### Yeast strain and growth conditions
*S. cerevisiae* strain BY4741 was grown in YPD media at 30 °C. Saturated cultures were diluted into 300 ml YPD medium with an optical density at 660 nm (OD660) of approximately 0.1 and then grown at 30°C with shaking at 250 rpm until mid-log phase with an OD660 of 0.6-0.7 at the time of NAI-N₃ treatment.

### In vivo NAI-N₃ modification of RNA
For in vivo modification by NAI-N₃, 50 ml of exponentially growing yeast cells at 30 °C were harvested and resuspended in 475 μl modification buffer (467.5 μl 1× PBS, 5 μl 1 mg/ml *Actinomycin D* (Sigma Aldrich, Cat. A1410), 2.5 μl 10% SDS). Note that for icSHAPE library construction (mature RNA), *Actinomycin D* was removed from the modification buffer. The samples were then incubated with either 25 μl of 2 M NAI-N₃ in DMSO (in vivo modification) or 25 μl of pure DMSO (DMSO control) with moderate shaking (10 rpm) at 30 °C for 10 min, centrifuged and washed twice with 1× PBS to remove traces of NAI-N₃. The pellets were stored at −80 °C until further use.

Note that we used *Actinomycin D* to stop further transcription during NAI-N₃ treatment. Although, direct interaction between *Actinomycin D* and RNA or DNA:RNA hybrid duplex is unlikely[47] due to the steric hindrance between the 2-amino group of the phenoxazone ring and the 2′-hydroxyl group of RNA, indirect interference of the nascent RNA structure via competitive bindings with the single-stranded DNA[48] is still possible. To show that our result would not be significantly altered by the usage of actinomycin D, we extracted the sequence motifs known to bind to *Actinomycin D*[48] in the genes captured by eSPET-seq data. We found that only 32 genes contain these motifs, such that excluding these genes/regions did not alter the direction or statistical significance of any relevant correlations between nascent folding, R-loop and mutation rate. We therefore concluded that the effect of *Actinomycin D* is largely negligible across our study.

### In vitro NAI-N₃ modification of RNA
RNA was extracted from cells treated with DMSO. One micrograms RNA was resuspended in 5.2 μl RNase-free water, and then denatured at 95 °C for 2 min. It was quickly transferred to ice to cool, refolded in SHAPE reaction buffer at 30 °C for 5 min by adding 3.3 μl 3.3× SHAPE buffer (333 mM HEPES (pH 7.5), 20 mM MgCl2 and 333 mM NaCl) and 0.5 μl RiboLock (Invitrogen, EO0384), followed by NAI-N₃ modification for 10 min at 30 °C by adding 1 μl 1 M NAI-N₃. The modified RNA was cleaned up and eluted in RNase-free water by using RNA Clean&Concentrator™−5 kit (Zymo, R1014).

### Cell fractionation for nascent RNA extraction
*S. cerevisiae* fractionation was performed as previously described[49]. Specifically, each yeast pellet from a 50 ml culture (OD660-0.7) was homogeneously suspended in 1.5 ml buffer Y1 (1 M sorbitol, and 0.1 M EDTA at pH 7.4), with 50 μl lyticase (Sigma Aldrich, L2524, 10 U/μl in Y1 buffer), and the samples were incubated at 30°C with rotation (100 rpm) until >90% of the cells became spheroplasts (-30 min). Spheroplasts were collected by centrifugation at 300 × g for 5 min at 4°C, washed twice with SB buffer (1 M sorbitol and 20 mM Tris-Cl at pH 7.4) and resuspended in 500 μl precooled cytoplasmic lysis buffer

(20 mM Tris-Cl at pH 7.4, 100 mM NaCl, 0.5% Triton X-100 (SIGMA), 15 mM β-ME, 50 U/ml SUPERase•In™ RNase Inhibitor (Invitrogen, 00795321) and 1× Complete EDTA-free protease inhibitor (Roche,04693132001)) to lyse the cytoplasmic membrane, and the samples were kept on ice for 10 min with gentle mixing every 3 min. The lysate was layered over 1 ml sucrose buffer (20 mM Tris-Cl at pH 7.4, 100 mM NaCl, 1.2 M sucrose, 15 mM β-ME, 50 U/ml SUPERase•In™ RNase Inhibitor and 1× Complete EDTA-free protease inhibitor) and centrifuged at 13,000 g for 15 min at 4 °C. The supernatant contained the cytoplasmic fraction (Cytoplasm). An aliquot (50 µl) was taken for Western analysis and RT-qPCR, and the rest of the supernatant was discarded. The white nuclear pellet was resuspended in 500 µl nuclei lysis buffer (20 mM Tris-Cl at pH 7.4, 100 mM NaCl, 1.0% Triton X-100, 15 mM β-ME, 50 U/ml SUPERase•In™ RNase Inhibitor and 1× Complete EDTA-free protease inhibitor) to lyse the nuclear membrane. Samples were kept on ice for 10 min with max speed vortex for 5 s every 3 min and the chromatin fraction was collected by centrifugation at 16,000 × $g$ for 10 min at 4 °C. Chromatin pellets were washed three times with fractionation wash buffer (0.1% Triton X-100, 1 mM EDTA and 1× PBS) and suspended in 250 µl 1× PBS. An aliquot (50 µl) was taken for Western analysis and RT-qPCR (Chromatin). The remaining chromatin solution was stored at −80 °C until further use.

To quantify the nascent RNA enrichment in cell fractionation, RNAs were isolated from the chromatin (nascent RNA) and cytoplasmic (mature RNA) fractions using the miRNeasy Mini Kit (Qiagen, 217004) according to the manufacturer's instructions. One hundred nanograms of RNA was reverse transcribed using SuperScript III reverse transcriptase (Invitrogen,18080044) with specific primers (Supplementary Fig. 1b, Supplementary Table 2), following the manufacturer's protocol. We selected *ADH1* as a molecular indicator to estimate the extent of nascent RNA enrichment. Because nascent RNAs contain the regions after the polyadenylation sites that are present in the nascent transcripts but absent from the mature mRNAs, *ADH1* nascent RNA was selectively detected by priming an RT reaction downstream of this site[50]. Mature *ADH1* transcripts were detected by priming an RT reaction at the poly(A)-tail with oligo(dT) primer[50]. cDNAs were quantified by qPCR to yield the abundance of mRNA and nascent RNA in the samples. Enrichment was calculated by the concentration ratio between the nascent RNA and the mature RNA from the chromatin fractions, normalized to the same ratio from the cytoplasmic fractions.

Western blots of different samples from yeast cell fractionation were carried out to assay the success of the fractionation. Western blotting was performed as previously reported with minor modifications[51]. Briefly, protein samples from different fractions were mixed with 5× protein loading buffer, boiled at 95 °C for 5 min and then resolved by SDS-PAGE. Next, separated proteins were transferred to nitrocellulose membranes and probed with the following primary antibodies against chromatin proteins (histone H2B, Abcam, ab188291,1:2000 dilution) and cytoplasmic proteins (tubulin, Abcam, ab185224,1:5000 dilution). Finally, the membrane was probed with horseradish peroxidase (HRP)-conjugated goat anti-rabbit IgG (Abcam, ab6721,1:10,000 dilution) and scanned using a ChemiDoc Touch Imaging System (BioRad).

### Nascent RNA extraction from chromatin fraction

The chromatin solution (200 µl) containing nascent RNAs was resuspended in 0.8 ml TRIzol™ LS reagent (Invitrogen, 10296010CN), thoroughly mixed and incubated at 50 °C for 5 min. Then, 200 µl chloroform was added and shaken vigorously for 15 s. After centrifugation at 12,000 × $g$ for 15 min at 4 °C, the aqueous phase was transferred to a 1.5 ml RNase-free tube and purified using the miRNeasy Mini Kit (Qiagen, 217004) and on-column DNase I digestion was performed using the RNase-free DNase set (Qiagen, 79254) according to the manufacturer's instructions. For each treatment, three

independent biological replicates were performed for the nascent RNA isolation and library construction.

### Preparation of eSPET-seq library of nascent RNA at the genome scale

After nascent RNA extraction, sequencing libraries were prepared similarly to previous protocols used for icSHAPE[23] and Structure-seq[52], with some modifications. The whole process can be divided into six steps, namely biotin click reaction of NAI-N$_3$, 3′ adapter ligation, removal of excessive 3′ adapter, cDNA synthesis and isolation of NAI-N$_3$-modified molecules, 5′ adapter ligation, and library amplification by PCR. Each individual step is described in detail below.

Step one is the biotin click reaction of NAI-N$_3$. To specifically select NAI-N$_3$-modified RNA, all nascent RNA samples were treated by a biotin-alkyne click reaction (Click-iT™ Biotin sDIBO Alkyne, Invitrogen, C20023,). Note that DMSO-treated RNA samples were not capable of 'click' reaction due to their lack of the NAI-N$_3$ modification, but this experimental step was still carried out for the DMSO-treated control sample in parallel with the NAI-N$_3$-treated samples. For each reaction, 3 µg of nascent RNA was used in a final volume of 100 µl by adding 100U RiboLock, 37 µM 1.85 mM DIBO-biotin, 50 µl 2× PBS. Biotinylated RNAs were purified using the Zymo RNA Clean & Concentrator-5 column according to the manufacturer's protocol and eluted in 7 µl of RNase-free water.

Step two is the ligation of 3′ adapters. Due to a lack of a biotin moiety for the DMSO control, DMSO-treated samples require a 3′ biotinylated RNA adapter (Supplementary Table 2), whereas NAI-N$_3$-treated samples require a 3′ ddC RNA adapter (Supplementary Table 2). A total of 100 pmol of a 3′-adapter was ligated to the biotinylated nascent RNA in a reaction volume of 20 µl by adding 2 µl of 10× RNA ligase buffer (NEB), 1 µl of T4 RNL2tr K227Q (NEB), l µl of SUPERase•In™ RNase Inhibitor, 2 µl of 100% DMSO and 6 µl of 50% PEG8000. Each sample was gently mixed and then incubated at 37 °C for 3 h. The 3′ end ligated nascent RNA was purified using the Zymo RNA Clean & Concentrator-5 column according to the manufacturer's protocol and eluted in 30 µl of RNase-free water.

Step three is the removal of excessive 3′ adapters. After the ligation reaction, the excessive 3′ adapter was degraded with RecJ exonuclease after deadenylation of the 3′ adapter with 5′ deadenylase. Briefly, 2 µl of 5′ deadenylase(NEB,M0331S), 2 µl of RecJ(NEB, M0264S), 4 µl of 10× RecJ buffer and 2 µl of SUPERase•In™ RNase Inhibitor were added to the product of the ligation reaction above. The samples were then incubated at 37 °C for 1 h. Reactions were stopped by purification with the Zymo RNA Clean & Concentrator-5 column according to the manufacturer's protocol, and the samples were then eluted in 11 µl of RNase-free water.

Step four is cDNA synthesis and isolation of NAI-N$_3$-modified molecules. RT primer (1 µl, 50 µM) (Supplementary Table 2) was added to the 3′ adapter-ligated nascent RNA samples. The samples were heat-denatured at 70 °C for 5 min, then cooled slowly to 25 °C (1 °C per s) and then incubated at 25°C for 5 min. After primer annealing, cDNA synthesis was carried out in a final volume of 20 µl (containing 0.5 mM dNTPs, 50 pmol RT primer, 5 mM DTT, 1× First-Strand Buffer, 20 U SUPERase•In™ RNase Inhibitor and 200 U SuperScript® III Reverse Transcriptase(Invitrogen)) by incubation at 25°C for 3 min, 42°C for 5 min, and 52 °C for 30 min. Enrichment of NAI-N$_3$-modified molecules was performed as previously described[23]. NAI-N$_3$-enriched cDNAs were purified using the Zymo Oligo Clean & Concentrator column (Zymo, D4061) according to the manufacturer's protocol and eluted in 6 µl of RNase-free water.

Step five is the ligation of the 5′ adapter. The ligation was performed using T4 RNA ligase 1 (NEB, M0437M) which ligated the 3′ end of the cDNA to a single-stranded DNA adapter (where "N" denotes a mixture of all four bases, and the three "N" were used to distinguish PCR duplicates created by step six below). The reaction was carried out

in a final volume of 20 µl, in the presence of 1× RNA ligase buffer, 20% PEG8000, 1 mM ATP, 100 pmol DNA adapter, and 30 U T4 RNA ligase. The ligation was performed at 25 °C for 16 h and then deactivated at 65 °C for 15 min. The 5′ adapter-ligated cDNA was then purified using Oligo Clean & Concentrator columns according to the manufacturer's protocol, and eluted in 20 µl of RNase-free water.

Step six is the library amplification and purification. PCR amplification was performed on the ligated cDNA using Q5 High Fidelity DNA polymerase (NEB, M0530) and Illumina TruSeq primers. For each experiment, we took 5 µl of the ligated cDNA and set up an 80 µl reaction for PCR cycle optimization. Reactions (80 µl) were performed with 1× Phusion HF buffer, 0.2 mM dNTPs (each), 0.8125 µM forward primer, 0.5 µM reverse primer (Supplementary Table 2, in which the NNNNNN was the multiplexing index) and 1.6 U Phusion DNA polymerase. To determine the minimum number of cycles required to obtain sufficient products, we set up four PCR tube strips and transferred a 20 µl aliquot of the PCR mixture into one tube in each strip. PCR amplifications were carried out with varying numbers of cycles (10, 12, 14, and 16) by placing all strip tubes in the thermal cycler and starting a program with the following conditions: 98 °C for 1 min, 98 °C for 15 s, 60 °C for 30 s, 72 °C for 60 s. For the remaining ligated cDNAs (15 µl), the amplification was completed at the selected cycle number. The PCR product was then run on a 10% native polyacrylamide gel to remove the byproduct, and the slice corresponding to 150–600 nt was cut using both a 50-bp DNA Ladder (Takara, 3421) and a Low Range DNA Ladder (Invitrogen, SM1193) as references. The DNA in the slice was resolved by passive diffusion in diffusion buffer for 16 h at 37 °C in a Thermomixer at 1000 rpm and purified from the residual PAGE gel using a 0.45 µm Spin-X columns (Corning, 431481) and the Zymo DNA Clean & Concentrator-5 column (Zymo, D4014). The dsDNA libraries were then sequenced with 2 × 150 paired-end reads on an Illumina HiSeq X Ten. All sequenced samples and their summary statistics were listed in Supplementary Table 1.

### Preparation of icSHAPE library of mature RNA

icSHAPE sequencing libraries were prepared as follows. First, total RNA extraction, isolation of mRNAs, biotin click reaction of NAI-N$_3$, RNA fragmentation, RNA end repair and RNA 3′-end ligation were conducted as previously described[18]. The RNAs were then purified and eluted in 31 µl RNase-free water by using RNA Clean&Concentrator™-5 kit. The purified RNA was incubated in a 40 µl reaction by adding 4 µl 10× NEB buffer 2, 2 µl RecJ, 2 µl 5′Deadenylase,1 µl Ribolock at 37 °C for 60 min. Then reverse transcription and biotin-streptavadin enrichment was performed as described in the icSHAPE assay[18]. The enriched first-strand cDNAs were then ligated at their 3′ ends to a ssDNA linker, and Illumina sequencing adapters and indexes were introduced by 8–14 cycles of PCR using Phusion HF Polymerase as described above in our eSPET-seq procedure. The dsDNA libraries were then sequenced with 2 × 150 paired-end reads on an Illumina NovaSeq 6000. All sequenced samples and their summary statistics were listed in Supplementary Table 1.

### Inference of prevalence of secondary structures by eSPET-seq or icSHAPE data

Raw sequencing reads were first trimmed to remove the 3′ adapters by Trimmomatic[53] (V 0.38). The three nucleotides corresponding to the three "N" in the 5′ adapter (see step five in the eSPET-seq library preparation above) were also removed by cutadapt[54] (V 1.18). Any reads shorter than 18 nts were discarded. The *S. cerevisiae* genome and annotation (R64-1-1) were downloaded from EnsEMBL. The processed forward (corresponding to NAI-N$_3$-modified site) and reverse reads (corresponding to transcription site in eSPET-seq and random DNA fragmentation in icSHAPE) were separately mapped to the genomic sequences (intron retained) of all genes using Bowtie2 with the parameters "−norc" for forward reads, "−nofw" for reverse reads.

Any reads with a mismatch on the first nucleotide at the 5′ end or mapping quality score <30 were discarded. Reads with ambiguous origins, or likely PCR duplicates (judged from the mapped position of both reads and the three "N") were further removed. Biological replicates, which were highly correlated with each other (Fig. 1b), were pooled together unless otherwise stated.

To investigate cotranscriptional folding using eSPET-seq data, we collected all read pairs from whose reverse reads were mapped to the same position, which therefore corresponded to the same transcriptional intermediate. The base immediately preceding the first 5′ end mapping position of the forward read was considered the RT stop site caused by the NAI-N$_3$ modification. For each transcriptional intermediate, the sum of RT stops at each position was calculated, and the density of RT stops was normalized within each transcriptional intermediate by 90% winsorization. For testing structural transitions during cotranscriptional folding, a sliding window with a length of 50 nt was used to scan the longitudinal profile (Fig. 2c) of RT stop density for each nucleotide. Only windows with at least one read in the first half of the window and at least one read in the second half of the window were considered. Structural transitions were identified as windows whose first and second halves have significantly different densities of RT stops by the Wilcoxon test. Note that the DMSO-treated samples were not involved here because DMSO-treated samples were used to control NAI-N$_3$-independent RT stops due to sequence features (e.g., GC%). Such NAI-N$_3$-independent RT stops should be constant for the same nucleotide; therefore, longitudinal comparisons made here for the same nucleotide between transcriptional intermediates intrinsically controlled the probability of NAI-N$_3$-independent RT stops. For all other analyses, we used a single-stranded score, which controlled the probability of NAI-N$_3$-independent RT stops by contrasting the NAI-N$_3$ samples with the DMSO samples (see below).

For both eSPET-seq and icSHAPE data, a single-stranded score for each nucleotide was estimated based on the NAI-N$_3$ reactivity reflected by the forward reads, following previous methods[55]. Briefly, the number of RT stops was normalized to take into account the different sequencing depths between the NAI-N$_3$ and DMSO samples, using the normalization constants $k_{DMSO}$ and $k_{NAI-N3}$, which are defined as follows:

$$k_{DMSO} = \frac{S_{DMSO} + S_{NAI-N3}}{2 \times S_{DMSO}} \tag{1}$$

$$k_{NAI-N3} = \frac{S_{DMSO} + S_{NAI-N3}}{2 \times S_{NAI-N3}} \tag{2}$$

$S_{DMSO}$ and $S_{NAI-N3}$ are the total number of mapped reads in the DMSO- and NAI-N$_3$-treated samples, respectively. Then, the normalized number of RT stops for nucleotide $i$ of a gene in the DMSO and in vivo samples were calculated as:

$$N_{DMSO}(i) = k_{DMSO} \times n_{DMSO}(i) \tag{3}$$

$$N_{NAI-N3}(i) = k_{NAI-N3} \times n_{NAI-N3}(i) \tag{4}$$

where $n_{DMSO}(i)$ and $n_{NAI-N3}(i)$ are the raw numbers of RT stops for position $i$ of a gene in the DMSO and NAI-N$_3$ samples, respectively. Finally, the single-stranded score of nucleotide $i$ was calculated as:

$$\theta_{NAI-N3}(i) = \log_2\left(\frac{N_{NAI-N3}(i) + 1}{N_{DMSO}(i) + 1} + 1\right) \tag{5}$$

Note that pseudocounts of 1 were added to avoid a logarithm of or division by zero and made the final score larger than zero. The final

single-stranded score is larger than 0 and not larger than 7, with any raw $\theta_{NAI-N3}$ larger than 7 capped at 7.

At the gene level, the single-stranded score of a gene was calculated as the arithmetic mean of the single-stranded scores of all the nucleotides within the gene. We also calculated the Gini index (by R package "ineq") using the single-stranded scores of all nucleotides within a gene to represent the average prevalence of RNA secondary structure of the gene, as it has previously been shown that as the structure unfolds, the single-stranded score becomes more even (low Gini index)[5].

### Verification of structural transition-coupled R-loop dissolution on 5S RNA

A DNA:RNA immunoprecipitation (DRIP. See below) followed by reverse transcription quantitative PCR was conducted to determine whether nucleotides 30–45 of 5S rRNA form an R-loop. Briefly, DRIP was first used to purify R-loops by utilizing the high specificity and affinity of the S9.6 monoclonal antibody. Reverse transcription was conducted using three different RT primers designed for transcription intermediates L1, L2 and L3 (length = 78 nt, 105 nt and 115 nt, respectively. Supplementary Fig. 4a). The resulting cDNA was quantified by qPCR with primers targeting nucleotides 30–45 of the 5S rRNA, and then normalized to *ACT1* to yield the relative R-loop signal in this region (Supplementary Fig. 4a). We found that the R-loop signal is significantly higher for intermediate L1 than L2 and L3 (Supplementary Fig. 4b, three bars on the left). To further verify that this observed difference is indeed caused by R-loop formation, we overexpressed *RNASEH1* (it hampers R-loop formation by degrading the RNA in R-loop) and repeated the same experiment of DRIP followed by RT-qPCR (Supplementary Fig. 4b, three bars on the right).

### Detection of R-loop by DRIP-RT-qPCR

In order to detect R-loop in specific regions, we performed DNA:RNA immunoprecipitation followed by reverse transcription quantitative PCR(DRIP-RT-qPCR) as previously[10] described with some minor modification. Briefly, exponentially growing BY4741 and its mutant derivatives (OD600 = 0.7, 10 ml) were crosslinked for 25 minutes at room temperature using 1% formaldehyde, and quenched for 5 min using 360 mM glycine(Sigma, 50046). The pellets were rinsed with PBS for 5 min, resuspended in 400 µl FA-1 lysis buffer, mixed with 200 µl of glass beads (Sigma,18406), and vortexed for 15 min at 0 °C in a Thermomixer at full speed. Glass beads were removed after transient centrifugation. By centrifuging at full speed for 15 min at 4 °C, the cross-linked chromatin was recovered, and then resuspended in 800 µl of FA-1 buffer, and sonicated to obtain DNA fragments of approximately 150 bp. Sonicated chromatins were centrifuged at 20,000 × *g* for 15 min at 4 °C to remove cellular debris, and 5% glycerol (Sigma, G5516) was added to the supernatant. The supernatants were mixed with 25 µg of antibody S9.6 (Kerafast, ENH001) and 100 µl of prewashed Protein A agarose (Sigma, P2545) and rotated overnight at 4 °C. We recovered the agarose beads, washed them successively with FA-1 buffer, FA-2 buffer, FA-3 buffer, and TE at 4 °C, then added 700 µl of QIAzol reagent and mixed them by vortexing. RNA was purified using the miRNeasy mini kit according to the manufacturer's instructions, including the optional on-column DNase treatment with the RNase-free DNase set (Qiagen, 79254). The purified RNA was then used in RT-qPCR to examine the relative R-loop signal. The primers used for RT-qPCR in this study are listed in Supplementary Table 2.

### Secondary structure models

Secondary structure models (Figs. 1f and 2b) were plotted by VARNA (http://varna.lri.fr/). The known structure of tRNA was extracted from RNAcentral (https://rnacentral.org) and that of 5S rRNA was extracted from RiboVision (http://apollo.chemistry.gatech.edu/RiboVision).

### Ratio of reads mapped to exons versus introns

The nascent RNA 3′-end sequencing data were downloaded from NCBI Sequence Read Archive under accession number SRR3177717. Initial analysis of the sequencing data was carried out in the same manner as previously reported[56], after which the "bigwig" files were created from BAM files using bamCoverage of deepTools[57] with default parameters. A custom script was used to extract intronic and exonic read counts for all intron-containing genes from the bigwig files according to the annotation of *S. cerevisiae* genome (R64-1-1). The total number of exonic and intronic reads per gene are normalized by their respective lengths. The ratio of the size-normalized counts is calculated for each gene containing at least one read in both exonic and intronic regions (Supplementary Fig. 1c).

### Expression levels and evolutionary conservation of *S. cerevisiae* genes

The expression level of the yeast transcriptome was extracted from a previous RNA-seq-based report[58]. Evolutionary conservation was estimated inversely by the ratio between the number of nonsynonymous substitutions per nonsynonymous site ($d_N$) and the number of synonymous substitutions per synonymous site ($d_S$) detected from one-to-one orthologs between *S. cerevisiae* and *Saccharomyces bayanus* following previously described pipelines[35].

### R-loop score

The yeast S1 nuclease DNA-RNA immunoprecipitation with deep sequencing (S1-DRIP-seq) data[29] were downloaded from NCBI Sequence Read Archive (SRA) under the accession number SRP071346 specific runs used were: SRR3504389-SRR3504390, SRR3504393-SRR3504394, and SRR3504396 (wild-type data and the corresponding input-chromatin control data). The procedure of the S1-DRIP-seq data analysis was the same as previously reported[29]. Let the number of reads whose 5′ most nucleotide mapped to a given site be *n*, and the sum of *n* over all sites in the genome be *m*. We calculated $x = \log_2((n + 1)/m)$, and the R-loop score of a site was defined by *x* calculated from the wild-type data subtracted by *x* calculated from the input-chromatin data for the site. Human R-loop score (R-ChIP) data were downloaded from NCBI Gene Expression Omnibus available under the accession number GSE97072. The longest transcript was selected when genes had more than one transcript.

### Mutation rates across the *S. cerevisiae* genome

All single nucleotide variations (SNVs) from previously compiled population genomic data of 190 *S. cerevisiae* strains, as well as their inferred phylogenetic relationship, were extracted from the original publication[28]. Based on this dataset, we applied the GAMMA algorithm[59] to estimate the (relative) mutation rate for each segregating site, as described below. First, based on the phylogeny and the genotypes of the focal site, GAMMA first inferred its ancestral state on each internal node as the one with the highest (posterior) probability using a distance-based method[60]. Second, GAMMA estimates the expected number of mutations per unit of time at the focal site using a maximum likelihood (ML) method, which takes into account the possibility of multiple mutations appearing as only one nucleotide change in a long branch. Specifically, let us denote the total branch length of the full phylogeny as *B*, and the total number of mutations happened on the focal site within the full phylogeny as *k*. On a branch *i* with length $b_i$, the number of mutations on the focal site follows a Poisson distribution with the expectation of $kb_i/B$. Thus, the probability of no change on branch *i* is $p_i = e^{-kb_i/B}$, and the probability of a change (which might be the result of more than one mutations) is $q_i = 1 - p_i = 1 - e^{-kb_i/B}$. We can further divide all branches within the phylogeny into two groups: those that have undergone genotype changes (denoted by $G_1$), and those that have not (denoted by $G_0$). Then the likelihood of observing the empirical data can be given by

$L = \prod_{i \in G_1} q_i \prod_{j \in G_0} p_j = \prod_{i \in G_1}(1 - e^{-kb_i/B}) \prod_{j \in G_0} e^{-kb_i/B}$. Finally, GAMMA estimates $k$ by finding a positive solution of equation $\partial \ln L / \partial k = 0$, since $L$ is maximized (i.e. maximum likelihood) when the derivative of $\ln L$ is 0. As a result, each segregating site has its own ML-estimation of $k$, which is then divided by $B$ and used as an approximation of per-site mutation rate (relative to other sites within the genome). Note that the unit of time for $k$ is $B$, which is shared by all sites since they share the same full phylogeny. As a result, $k$ values from different sites can be compared regardless of the actual value of $B$. Non-segregating sites were assumed to have a zero (negligible) mutation rate. Non-segregating sites were assumed to have a zero (negligible) mutation rate.

It is imperative to note that using DNA polymorphisms to estimate mutation rates is not ideal. There is a possibility that the observable level of DNA polymorphism, or mutation rate thereby estimated, may be affected by both mutation rate and natural selection, especially negative/purifying selection, as positive selection is generally much rarer. However, according to the neutral theory of molecular evolution[61], most intraspecific polymorphisms are selectively neutral[62], which is indeed empirically supported[63,64]. For example, it was previously estimated that only 12% of coding SNPs found in yeast populations are deleterious[64]. Using Tajima's $D$ test[65], we also independently validated such general neutrality of polymorphisms in the current dataset of 190 strains by confirming that polymorphisms in the whole genome ($P = 0.1$ by Tajima's $D$ test) and the vast majority of genes were indeed compatible with the neutral expectation (only 604 out of 6079 genes show a $P < 0.05$ by Tajima's $D$ test, and none of them show a $P < 0.05$ when corrected for multiple testing using the Benjamini-Hochberg Procedure). Such general neutrality of polymorphisms dictates that the majority of variation in polymorphism levels can be attributed to variations in mutation rates rather than natural selection. Indeed, a higher level of polymorphism in a particular genomic region has often been used as an indicator of a higher local mutation rate[66–68]. Moreover, we found that there is a positive correlation between our polymorphism-based mutation rates and those estimated from three previously published mutation accumulation datasets[30–32] (Spearman's $\rho = 0.52$, $P < 0.05$. Supplementary Fig. 10), which is generally accepted as the best direct estimate of mutation rate. Here the mutation rate of a gene was estimated by the total number of mutations identified in the gene divided by the gene length[10]. This correlation further supports our view that polymorphism-based estimates are reasonable approximations for the mutation rate relative to other sites within the genome. Most importantly, even when we used mutation rate estimates from mutation accumulation experiments, we are still able to detect the antimutator effect of nascent RNA structure (Supplementary Fig. 7).

To distinguish the independent effects of TCR and the antimutator effect of nascent RNA folding, we used mutation events detected in three mutation accumulation experiments[30–32] (Supplementary Fig. 9). Here mutation accumulation experiments instead of the 190 *S. cerevisiae* strains were used because we need the mutational direction. Assuming dominate contribution by hydrolytic deamination of cytosine in C/G to T/A mutations[44], we assigned all C-to-T mutations as happened on the Watson strand and all G-to-A mutations as C-to-T mutations happened on the Crick strand.

## Odds ratios

To assess within-gene correspondence among single-stranded scores (S), R-loop scores (R), and mutation rates (M), we calculated three odds ratios, namely, $OR_{S\text{-}R}$, $OR_{R\text{-}M}$, and $OR_{S\text{-}M}$. To calculate $OR_{S\text{-}R}$, a 2 × 2 table was constructed for each gene by respectively classifying each nucleotide into one of four groups on the basis of (i) whether its single-stranded score is higher than the mean single-stranded score within the gene and (ii) whether its R-loop score is higher than the mean R-loop score within the gene. Let the numbers of nucleotides that fall into the four categories be: $a$ (yes to both questions), $b$ (yes to only question i), $c$ (yes to only question ii) and $d$ (no to both questions), respectively. The number of nucleotides in each category was added by 1 as a pseudocount to avoid division by zero. Then, $OR_{S\text{-}R} = ad/bc$. If strong nascent RNA folding reduces the R-loop, $OR_{S\text{-}R}$ should be > 1. The function "mantelhaen.test" in R was used to combine the $OR_{S\text{-}R}$ values from all genes and perform the MH test (Cochran-Mantel-Haenszel chi-squared test). Similarly, to calculate $OR_{R\text{-}M}$, a 2 × 2 table was constructed for each gene by respectively classifying each nucleotide into one of four groups on the basis of (i) whether its R-loop score is higher than the mean R-loop score within the gene and (ii) whether its mutation rate is higher than the mean mutation rate within the gene. $OR_{R\text{-}M}$ should therefore be > 1 if a weak R-loop reduces the mutation rate. To calculate $OR_{S\text{-}M}$, a 2 × 2 table was constructed for each gene by respectively classifying each nucleotide into one of four groups on the basis of (i) whether its single-stranded score is higher than the mean single-stranded score within the gene and (ii) whether its mutation rate is higher than the mean mutation rate within the gene. $OR_{R\text{-}M}$ should therefore be > 1 if strong nascent RNA folding reduces the mutation rate.

## Manipulative experiment of *CAN1*

Our manipulative experiment focused on the *CAN1* gene, which encodes an arginine permease that is commonly used to detect spontaneous mutations. It has been demonstrated[10] that a functional *CAN1* is lethal in media containing the toxic arginine analog canavanine, whereas a *CAN1* null mutant is viable. By measuring the frequency of colonies that were resistant to canavanine, the spontaneous mutation rate in the *CAN1* gene can be estimated. Furthermore, S1-DRIP-seq data[29] (see above) demonstrated that *CAN1* forms R-loops in the first 300 bp of its 5' end during transcription (Fig. 5a), thereby enabling manipulative experiment.

Utilizing this reporter gene, we first determined whether increased nascent RNA folding would reduce spontaneous mutations. To manipulate nascent RNA folding strength around the region prone to form R-loop (the first 300 bp at 5'-end of *CAN1*), we first generated synonymous mutants of *CAN1* by shuffling all synonymous codons within this region, so that the protein sequence and the codon usage bias of these mutants remained identical to wild-type *CAN1*. We then predicted the nascent RNA folding strength of these *CAN1* mutants using DrTransformer[69]. Considering the potential error of in silico structure prediction, we synthesized several mutants with various nascent RNA folding strengths, inserted them into the yeast genome and experimentally measured their nascent RNA structures with eSPET-seq. The eSPET-seq experimental pipeline was same as described above, with the exception of an additional round of 16-cylcle PCR amplification enriching *CAN1*-derived transcript intermediates (primers listed in Supplementary Table 2) before PAGE-purification and NovaSeq library preparation. Based on these *CAN1*-specific eSPET-seq results, three *CAN1* mutants with strong, intermediate and weak nascent RNA folding (Fig. 5b) were selected and tested for the corresponding spontaneous mutation rate.

To measure mutation rates of *CAN1*, we performed forward-mutation assay as described previously[10,70] with some minor modification. Briefly, for each strain (with different versions of *CAN1*), a single colony was grown at 30 °C in 5 ml synthetic complete (SC) medium overnight. The cells were diluted to ~100 cells/ml in 10 separate cultures for each strain and again grown to $1 \times 10^8$ cells/ml in SC medium. About ~$1.0 \times 10^4$ cells were plated onto canavanine-containing SC-arg plates (medium without arginine) to identify forward mutations in *CAN1* and onto SC plates to count the number of viable cells. Colonies appearing after 3 or 4 days of growth at 30° were counted. The spontaneous mutation rate of a strain was estimated by the number of $CAN^R$ colonies divided by the total number of cells after the growth in SC.

To further verify that the reduction of mutation rate by nascent RNA folding is mediated by inhibiting R-loop formation, we examined the prevalence of R-loops at the 5′ end of these three *CAN1* mutants by DRIP-RT-qPCR (see above). Consistent with our model, we found that the R-loop signal was significantly reduced when there was strong nascent RNA folding (Fig. 5d). To directly confirm that the antimutator effect of nascent RNA folding is R-loop-dependent, we stably over-expressed *RNASEH1* in the yeast genome, which hampers R-loop formation by degrading the RNA in an R-loop. It is predicted that *RNASEH1* overexpression reduces or even abolishes the effects of nascent RNA folding on spontaneous mutation rates, and this is indeed observed (compare Fig. 5c, e). Specifically, the significant reduction in mutation rate of *CAN1* sequence with intermediate folding compared to that with weak folding is no longer significant upon *RNASEH1* over-expression. Similarly, the antimutator effect of strongly-folded *CAN1* relative to the other two *CAN1* sequences was also weakened, although it remained statistically significant. More importantly, such a disappearance/weakening of the antimutator effect of nascent RNA folding was accompanied by the disappearance/weakening of R-loop prevalence reduction upon *RNASEH1* overexpression (contrasting Fig. 5d, f).

### Estimation of fitness advantage conferred by stronger nascent RNA folding

We estimated the fitness advantage conferred by the TAM-mitigation effect of stronger nascent RNA folding following our previous framework[10]. Briefly, assuming no recombination between the mutation target site and the corresponding site of the nascent RNA structure, the fitness advantage ($s$) conferred by a mutation that increases nascent RNA folding is approximately the reduction in the deleterious mutation rate ($\Delta\mu_d$) of its target site[40,71]. Let us assume that 70% of mutations at the target site are deleterious and that a mutation alters local nascent RNA folding can affect the mutation rate of a target of 10 sites. If the strengthened nascent RNA folding reduces the mutation rate of the target site by $x$-fold and the genomic average mutation rate per generation per nucleotide (which has already been reduced due to nascent RNA folding) is $y$, the fitness advantage can then be calculated as $s = \triangle u_d = 0.7 \times 10(x-1)y$. A phase diagram for the numerical relationship between $x$, $y$ and $s$ is shown in Fig. 6f. As a reference, the selective coefficient value that is equivalent to the inversions of the yeast effective population size is marked by a solid line. Therefore, the parameter space above the line represents a selective advantage large enough as a subject of natural selection (i.e., adaptive mutations), the parameter space below the line represents neutral mutations.

We attempted to estimate realistic values of $x$ and $y$ of the phase diagram. For $x$, we used 113 genes with at least 5% sites with mutation rate data and calculated the ratio between the average mutation rate of the 50-bp segment with the highest average single-stranded score, and that of the 50-bp segment with the lowest average single-stranded score. Infinite numbers (there are four of them) caused by division-by-zero were capped at 50. The resulting ratios for each gene appeared positively correlated with the gene expression level (Fig. 6e), suggesting that the mutation rates of the highly expressed (more functionally constrained) genes were more strongly reduced by nascent RNA folding. For $y$, it has previously been estimated[71] that the genomic average of per-generation per-nucleotide mutation rate in yeast is $3.3 \times 10^{-10}$. Two yeast genes, as indicated by the cross and the triangle, were marked on the phased diagram using the estimated $x$ and $y$ values.

### icSHAPE data for human nascent RNA

NAI-N$_3$ reactivity data for human nascent RNA by icSHAPE experiments were collected from a previous report[6]. The longest transcript was selected when genes had more than one transcript.

### Estimation of tumor mutation load

Gene-level somatic mutation data of the TCGA were downloaded from the UCSC Xena website (https://tcga.xenahubs.net), which consists of 37 types of cancer, with 36–9104 samples per type. We calculated the probability of each gene being mutated (regardless of the type of mutation) in all samples within each type of cancer. Then, the probabilities from all types of cancer were averaged and compared with the prevalence of nascent RNA folding. For the density of mutations, we downloaded the Mutation Annotation Files (MAF) for 33 types of cancer from the GDC data portal (https://portal.gdc.cancer.gov/). Next, the total number of mutations (regardless of cancer type) that appeared in each gene was divided by the gene length to calculate the mutation density.

### Reporting summary

Further information on research design is available in the Nature Portfolio Reporting Summary linked to this article.

## Data availability

The sequencing data generated in this study, i.e., eSPET-seq data, were deposited to NCBI Sequence Read Archive (SRA) under accession number SRP291653. Publicly available datasets used in this study were retrieved for yeast nascent RNA 3′-end sequencing available under accession number SRR3177717 at SRA, yeast S1-DRIP-seq data under accession number SRP071346 at SRA, human R-ChIP data under accession number GSE97072 at NCBI's Genome Omnibus (GEO), icSHAPE data for human nascent RNA under accession number GSE117840 at NCBI's GEO. Furthermore, somatic mutations (https://tcga.xenahubs.net) and mutation annotations (https://portal.gdc.cancer.gov/) in cancer, and tRNA sequences and secondary structure models (https://rnacentral.org) were used. Source data are provided with this paper.

## Code availability

Custom R/Python/Perl codes, that were used in data analysis, are available on GitHub (https://github.com/GongwangYu/NascentRNAfolding) and Zenodo (https://zenodo.org/record/8271608)[72].

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

## Acknowledgements

This work was supported by the National Key R&D Program of China (grant number 2021YFA1302500 and 2021YFF1200904 to J.-R.Y.), the National Natural Science Foundation of China (31671320, 31871320, 32122022 and 81830103 to J.-R.Y. and 31771406 to X.C.), the Sun Yat-sen University Founded Program (2022_76220_B21127 to J.-R.Y.) the Science and Technology Planning Project of Guangdong Province, China (2014A030304053 to X.Z.), and the Guangdong Basic and Applied Basic Research Foundation (2022A1515110749 to G.Y.). We thank Jianzhi Zhang, Qiangfeng Cliff Zhang, Chung-I Wu for their comments on the manuscript. The results shown here are in part based upon data generated by the TCGA Research Network: https://www.cancer.gov/tcga.

## Author contributions

J.-R.Y. conceived the idea, and designed and supervised the study. G.Y., Y.L., S.D. and Z.W. conducted experiments and acquired data. G.Y., Y.L., Z.L., S.D., Z.W., X.Z., W.C., J.Y., X.C. and J.-R.Y. analyzed the data. G.Y. and J.-R.Y. wrote the paper with inputs from all the authors.

## Competing interests

The authors declare no competing interests.
