## [Peer Review File · Nature Communications]

Genome-wide probing of eukaryotic nascent RNA structure elucidates cotranscriptional folding and its antimutagenic effectREVIEWER COMMENTS

Reviewer #1 (Remarks to the Author):

The manuscript from Yu GW et al describes a new nascent RNA structure sequencing method named "eSPET-seq" that they developed to study nascent RNA structures in eukaryotes (in this case, yeast). They used NAI-N3 instead of DMS for structure probing, isolated RNA from the chromatin fraction, ligated a 3'end adapter, before following the icSHAPE protocol for library preparation and sequencing. The authors claim that because they ligated the 3'adapter, they are probing RNA structures at the nascent 3'end. They then showed some positive controls and global patterns with regards to this dataset and in particular examined the role of nascent RNA structure and DNA mutations. While the work is potentially interesting, much of the data is not convincing and this limits my excitement about this current version of the manuscript. Below are my major concerns:

- 1) What proportion of the eSPET-seqs fall on nascent versus fully processed (mature) RNAs? How many transcripts can the authors detect structure reliably while they are still being processed (without polyA tails)? How much sequencing depth do we need to be able to capture nascent transcripts?
- 2) In the 5S rRNA example, the authors used an absence of reactivity around bases 30-50 to support that these bases are forming R-loops. But this is very indirect evidence – can the authors perform some kind of direct R loop capture, through proximity ligation experiments to show the correlation between R loop formation and structure transition for 5S or other positive controls?
- 3) In Figure 3c, the reactivity plot for mature RNA in vivo looks almost completely flat from bases 100-300- is there a problem with sequencing depth such that the authors are unable to capture the RNA structure in vivo?
- 4) The authors performed in vitro and in vivo structure probing to determine the effect of RBP binding on RNA structure but many things are changing during the refolding process- the RNAs are presumably also denatured and renatured on top of being stripped off their RNA binding proteins. Can the authors do a better comparison to use a gentle process with phenol-chloroform to remove the proteins but not denature the RNA?
- 5) The authors probed nascent transcripts but actually barely described any of the structural features of the nascent transcripts- for transcripts with introns- how does intron structure versus exonic structure look like? How do the structure of transcripts that are still being processed versus those that have finished processing but are still on the chromatin different?
- 6) With regards to R-loop formation and DNA mutation rates, most of the trends are very weak. Can the authors show one example of an important nascent structure and its role in something?

Reviewer #2 (Remarks to the Author):

In this work, Yu and collaborators adapted the SPETseq method to study RNA cotranscriptional structure in *S. cerevisiae*. The new method, which the authors named eSPETseq for eukaryote SPETseq is improved with respect to the previous version by the use of NAI-N3-dependent modification of all four bases instead of DMS which could modify only A and C nucleotides, and enriched for the chromatin fraction. This improved method substantially confirmed in yeast the results previously shown in bacteria confirming that the folding is mainly due to intrinsic RNA properties. The authors go further proposing a mechanism by which the RNA cotranscriptional folding at the 5'end, competing with the R-loop, would be protective toward the mutagenesis originated on the single stranded DNA generated as a consequence of the R-loop.

The first part of the paper is solid and the technical improvement is substantial. However, it comes late as nowadays more refined methods have been developed for the analysis of RNA structures, which rely on mutations profiles and take also into account of the cellular RNA heterogeneity. The second part of the work, namely the model by which RNA structure would be protective on DNA

mutagenesis, is mainly speculative. More solid data on specific models should be performed to substantiate the proposed model.

Reviewer #3 (Remarks to the Author):

This paper has the dual function of examining the incidence and dynamics of mRNA folding in yeast, and of relating this to information on mutation rates.

The method that they introduce to quantify / localize RNA folding does seem to be an advance over previous methods. However, Figure 1 and the description of the method needs to be substantially simplified / clarified. I am not entirely up on this kind of methodology, but I'm also not entirely ignorant, but I found the description of how one goes from starting raw RNA to estimated positions of stems and loops quite impenetrable. If other reviewers more in the know on the methodology are happy with this aspect of the paper, then it alone is a plus.

Bigger issues arise with respect to the analyses of mutational vulnerability, and I'm not convinced this should be included.

In particular, on line 220 – measuring mutation rates with DNA polymorphism data does not appear to be trustworthy. This not only reflects mutation, but also selection on individual sites, and the latter through both direct selection on sites and through local reductions in effective population sizes owing to selection on linked sites. Not only do the authors ignore the functional significance of the site, which will influence the standing levels of variation in potentially very dramatic ways (assuming there is indeed selection on folding), but they set mutation rates equal to zero when no polymorphisms are observed (which is a significant issue with sampling error associated with only moderate numbers of sequenced genomes).

So, overall, I am not confident in this part of the analysis. To their credit, the authors conclude that the strength of selection is insufficient for selection to be playing a role in generating the supposed local variation in mutation rates, which was a relief to see. I'll note in addition that prior work with mutation-accumulation experiments in a diversity of species has failed to find a compelling link between transcription and mutation rates, so this needs to be cited as well.

Although these authors emphasize tam, transcription-associated mutagenesis, there is also a phenomenon called transcription-coupled repair, so it is unclear that the former effect dominates. They do seem to try to check this, although one of the references they cite does not seem to be a mutation-accumulation experiment, so the overall test seems not to be strong (and they do not discuss statistical significance). Here, there is also the issue of the rate of elongation of individual transcripts vs. the rate of overall transcript production (which is presumably driven by initiation events). MA experiments simply do not generate enough data to perform fine-scaled analyses, but perhaps the authors could do a simple test to whether binned regions with and without secondary structure have different mutation rates, factoring out the problem of expression level / rate.

An explanation / definition is needed for the Gini index. Generally, this index is used as a measure of uniformity, and it just wasn't clear what the authors were doing here.

Finally, there are many grammatical errors that I have not attempted to correct, and this will need to be fixed prior to publication.

Reviewer #1's Comments

Overall Comment

The manuscript from Yu GW et al describes a new nascent RNA structure sequencing method named “eSPET-seq” that they developed to study nascent RNA structures in eukaryotes (in this case, yeast). They used NAI-N3 instead of DMS for structure probing, isolated RNA from the chromatin fraction, ligated a 3' end adapter, before following the icSHAPE protocol for library preparation and sequencing. The authors claim that because they ligated the 3' adapter, they are probing RNA structures at the nascent 3' end. They then showed some positive controls and global patterns with regards to this dataset and in particular examined the role of nascent RNA structure and DNA mutations. While the work is potentially interesting, much of the data is not convincing and this limits my excitement about this current version of the manuscript. Below are my major concerns.

Response:

We would like to thank the reviewer for his/her overall positive comments of our work.

Comment 1

What proportion of the eSPET-seqs fall on nascent versus fully processed (mature) RNAs? How many transcripts can the authors detect structure reliably while they are still being processed (without polyA tails)? How much sequencing depth do we need to be able to capture nascent transcripts?

Response:

We would like to thank the reviewer for his thoughtful comments. Our answer to each of these questions will be given in turn.

What proportion of the eSPET-seqs fall on nascent versus fully processed (mature) RNAs?

As mentioned in the comment, the presence of a polyA tail (at least 10 adenosines long, as a practical criteria) distinguishes mature RNA from nascent RNA¹. Since eSPET-seq captures the 3'-end of the transcript (Read 2 in Figure 1a), it is possible to determine whether a sequenced fragment originates from mature RNA by the presence of at least ten consecutive A bases at its 3'-end. Following this analysis, we determined that in the NAI-N₃ and DMSO groups, only 0.002% and 0.001% of eSPET-seq reads were derived from mature RNAs, respectively.

To further verify that eSPET-seq reads are mainly derived from nascent RNA, we analyzed a previously published yeast nascent RNA sequencing dataset, which specifically performed subtraction of poly(A)+ RNAs from chromatin-bound RNAs² (referred to below as "Oesterreich-2016"). We calculated the size-normalized ratio of reads mapping to exons versus introns for each gene. As an indicator of nascent RNA enrichment, this ratio was compared between the Oesterreich-2016 and eSPET-seq datasets. There was no significant difference in the ratio between the two datasets (Figure R1, also added to the manuscript as Supplementary Fig. 1c). These results indicate that eSPET-seq reads are primarily derived from nascent RNAs.

Figure R1. For both datasets of Oesterreich-2016 and eSPET-seq, size-normalized ratios of reads mapping to exons versus introns were calculated. There was no significant difference between the two datasets, suggesting a similar level of

enrichment for nascent RNA in the two studies. *P*-values are based on Wilcoxon signed-rank tests.

How many transcripts can the authors detect structure reliably while they are still being processed (without polyA tails)?

As mentioned above, the poly(A)+RNAs were first removed by filtering out sequenced fragments with at least ten consecutive A bases at their 3' ends (start of Read 2). In previous high-throughput assessments of RNA structure³, reliable detection of structure was obtained with ≥ 1 average reverse transcriptase stop per nucleotide (stop/nt). Following this practice, we obtained 1728 transcripts with ≥ 1 stop/nt and therefore reliable secondary structures.

How much sequencing depth do we need to be able to capture nascent transcripts?

To answer this question, we re-sampled (up- or down-sampled) the reads from eSPET-seq with replacement and re-estimated the number of genes with ≥ 1 stop/nt (for reliable detection of structure. See above) with the re-sampled reads, whose result is plotted below as Figure R2. This figure is also added to the manuscript as Supplementary Fig. 1d.

Figure R2. The correspondence between sequencing depth and the number of genes with ≥ 1 stop/nt. The red point indicates the estimated number using the full

eSPET-seq dataset. Accordingly, the points to the left are inferred by down-sampling, while the points to the right are inferred by up-sampling. The error bars indicate standard deviation estimated from 100 re-sampling procedures.

Comment 2

In the 5S rRNA example, the authors used an absence of reactivity around bases 30-50 to support that these bases are forming R-loops. But this is very indirect evidence – can the authors perform some kind of direct R loop capture, through proximity ligation experiments to show the correlation between R loop formation and structure transition for 5S or other positive controls?

Response:

We thank the reviewer for this helpful suggestion regarding how to confirm the relationship between R loop formation and structure transition. Following the reviewer's suggestion, we performed DNA:RNA immunoprecipitation (DRIP) followed by reverse transcription quantitative PCR (RT-qPCR) to verify the existence of R-loops (Figure R3a). Briefly, DRIP was first used to purify R-loops by utilizing the high specificity and affinity of the S9.6 monoclonal antibody. Reverse transcription was conducted using two different RT primers designed for transcription intermediates L1 and L2 (length = 80nt and 120nt, respectively). The resulting cDNA was quantified by qPCR with primers targeting nucleotides 30 to 45 of the 5S rRNA, and then normalized to *ACT1* to yield the relative R-loop signal in this region. We found that the R-loop signal is significantly higher for intermediate L1 than L2 (Figure R3b, left bars). To further verify that this observed difference is indeed caused by R-loop formation, we overexpressed RNASEH1 (it hampers R-loop formation by degrading the RNA in R-loop) and repeated the same experiment of DRIP followed by RT-qPCR. As a result, the R-loop signals were reduced for both L1 and L2, and their difference became insignificant (Figure R3b, right bars), suggesting the previous

significant difference was indeed R-loop dependent. These data directly showed the existence of R-loop for L1 and the absence (or at least, reduction) of R-loop for L2. Together with the density of RT-stops from eSPET-seq shown in Figure 2a, our results clearly demonstrated the correlation between R-loop formation and structure transition for 5S rRNA. These results have now been added to our manuscript as Supplementary Fig. 4.

Figure R3. The structural transition in 5S RNA was correlated with the dissolution of an R-loop. (a) Schematic diagram depicting the detection of the R-loop signal at nucleotides 30 to 45 of 5S rRNA. The density of RT stops in eSPET-seq data (same as shown in Figure 2a) suggests a structural transition when transcription proceeds beyond the length of 80nt. To confirm the corresponding R-loop dissolution, we performed DNA:RNA immunoprecipitation (DRIP) to enrich the R-loops, followed by reverse transcription using RT-primers designed for transcription intermediates L₁ (length = 80 nt) and L₂ (length = 120 nt), and finally qPCR using primers targeting nucleotides 30 to 45 to obtain a relative R-loop signal (relative to *ACT1*). (b) Wild-type strains (the "WT" group) showed a significantly higher R-loop signal in L₁ than L₂, consistent with R-loop dissolution for transcripts longer than 80 nt. More importantly, when RNASEH1 (which hinders R-loop formation by degrading RNA) was overexpressed, both L₁ and L₂ R-loop signals decreased and their difference is no longer significant (the "RNASEH1" group), which suggests the previous

significant L1-L2 difference is indeed R-loop-dependent. Together with the density of RT-stops from eSPET-seq, our results demonstrated the correlation between R-loop and structure transition for 5S rRNA. P-values are based on a two-tailed t-test.

Comment 3

In Figure 3c, the reactivity plot for mature RNA *in vivo* looks almost completely flat from bases 100-300- is there a problem with sequencing depth such that the authors are unable to capture the RNA structure *in vivo*?

Response:

We thank the reviewer for the question. There is sufficient sequencing depth (average 2000 for the *in vivo* group) to estimate the single-stranded score of YPL250C (the gene shown in Figure 3c). In Figure R4 below, we plotted the read coverage of mature RNAs *in vivo* in both the NAI-N₃ and DMSO groups.

Figure R4. Snapshot of eSPET-seq read coverage on YPL250C.

As shown in Figure R4, the 100-300 region is frequently sequenced by both the NAI-N₃ and DMSO groups. Because the single-stranded score is derived from the contrast of reads between the NAI-N₃ and DMSO groups (see “Inference of prevalence of secondary structures by eSPET-seq or icSHAPE data” in the Method section of our manuscript), this region should appear to have a consistently low single-stranded score (i.e. flat).

Comment 4

The authors performed *in vitro* and *in vivo* structure probing to determine the effect of RBP binding on RNA structure but many things are changing during the refolding process- the RNAs are presumably also denatured and renatured on top of being stripped off their RNA binding proteins. Can the authors do a better comparison to use a gentle process with phenol-chloroform to remove the proteins but not denature the RNA?

Response:

We would like to thank the reviewer for this question and the very constructive suggestion. Please allow us to recapitulate the context in which we investigated the effect of RBP binding. Our original question is how and why nascent RNA structures are similar/different from that of mature RNAs of the same gene. The hypothesis is that if a structure is mostly cis-regulated (i.e. determined by the thermodynamics dictated by the nucleotide sequence itself), the structure similarity between nascent and mature RNA should be high because their nucleotide sequences are identical. Otherwise, if a structure is mostly trans-regulated (i.e. determined by trans-factors such as RNA binding proteins or other RNA molecules that might interfere with the cis-regulated structures), the structural similarity between nascent and mature RNA should be low. To assess this hypothesis, we reasoned that cis-regulated structures should also be highly similar between *in vivo* and *in vitro* RNAs, and showed (Figure 3d) that this *vivo-vitro* similarity (i.e. cis-regulated structures) indeed correlates with mature-nascent similarity (i.e. cis-regulation underlies mature-nascent similarity). Here, the *in vitro* structure is exactly what we need, because the trans-regulated structures would be altered during re-folding in the *in vitro* environment, where the trans-regulators are removed. To further this logic, we proposed that RNA binding protein could be one major type of trans-regulator for RNA structure, and therefore hypothesized that for RNAs with more RBP binding (i.e. their structures are more likely trans-regulated), the *vivo-vitro* similarity and mature-nascent similarity should be lower. This is indeed the case (Figure 3e and f), further supporting our hypothesis about the role of cis-regulation in determining the structural similarity between

nascent and mature RNA.

We agree with the reviewer that the *in vitro* structures as we previously assessed were quite different from *in vivo* structures because of the vastly different (re-)folding environment. But as discussed above, our original reasoning actually relied on this very difference to probe the cis-regulated structures. Nevertheless, following the reviewer's suggestion, we conducted additional experiments. Briefly, we extracted native deproteinized total RNAs by a more gentle phenol-chloroform-based method under conditions previously shown⁴⁻⁷ to preserve native RNA structure, and then performed the standard icSHAPE experiment to probe the mRNA secondary structure (referred to as the "*ex vivo*" sample). As a result, we found that the structural similarity between *in vivo* and *ex vivo* mature RNA is lower for genes with more RBP binding (Figure R5). Assuming the structural similarity between *in vivo* and *ex vivo* mature RNAs is also mostly cis-regulated, this result again supports the role of cis-regulation (the nucleotide sequence) in RNA structures conserved across conditions (nascent/mature and *in vivo/ex vivo/in vitro*).

Figure R5. Standard boxplot showing that the top 50% of genes with higher structural similarity between mature RNAs folded *in vivo* and *ex vivo* tended to have less RBP binding sites than the other genes. *P*-value is from the Mann-Whitney U test.

Comment 5

The authors probed nascent transcripts but actually barely described any of the structural features of the nascent transcripts- for transcripts with introns- how does intron structure versus exonic structure look like? How do the structure of transcripts that are still being processed versus those that have finished processing but are still on the chromatin different?

Response:

We thank the reviewer for this thoughtful comment. As suggested by the reviewer, we have extended our manuscript to include a comparison of the structures of the intronic regions versus the exonic regions. It was found that RNAs in vivo are more folded in exonic regions than in intronic regions (median icSHAPE reactivity of 1.04 versus 1.12; a higher icSHAPE reactivity indicates a less structured region) (Figure R6, this figure is also added to the manuscript as Supplementary Fig. 5c and d).

Figure R6. Boxplot of negative of single-stranded score (c) and Gini index of single-stranded score (d) in exon versus in intron for intron-containing genes. P-values are based on Wilcoxon signed-rank tests. The exons showed stronger nascent RNA folding than the introns according to both metrics.

In regards to the second question, less than 0.01% of the reads in our eSPET-seq

data are derived from transcripts that have finished processing but are still on the chromatin (see our response to Comment 1 of Reviewer #1). As a result, we cannot make such a comparison with transcripts that are still being processed using our data. However, we intend to investigate this question in the future, which, however, should be included as a separate study since it is not highly relevant to the current study.

Comment 6

With regards to R-loop formation and DNA mutation rates, most of the trends are very weak. Can the authors show one example of an important nascent structure and its role in something?

Response:

We thank the reviewer for this critical suggestion. Per the reviewer's suggestion, we performed additional manipulative experiments to demonstrate the relationship between nascent RNA folding, R-loop and mutation rate for one particular gene, which is detailed below.

Our manipulative experiment focused on the *CAN1* gene, which encodes an arginine permease that is commonly used to detect spontaneous mutations. It has been demonstrated⁸ that a functional *CAN1* is lethal in media containing the toxic arginine analog canavanine, whereas a *CAN1* null mutant is viable. By measuring the frequency of colonies that were resistant to canavanine, the spontaneous mutation rate in the *CAN1* gene can be estimated. Furthermore, S1-DRIP-seq data demonstrated that *CAN1* forms R-loops in the first 300 bp of its 5' end during transcription (Figure R7a), thereby enabling manipulative experiment.

Utilizing this reporter gene, we first determined whether increased nascent RNA folding would reduce spontaneous mutations. To manipulate nascent RNA folding strength around the region prone to form R-loop (the first 300 bp of *CAN1*'s 5' end), we first generated synonymous mutants of *CAN1* by shuffling all synonymous codons within this region, so that the protein sequence and the codon usage bias of these

mutants remained identical to wild-type *CAN1*. We then predicted the nascent RNA folding strength of these *CAN1* mutants using DrTransformer⁹. Considering the potential error of *in silico* structure prediction, we synthesized several mutants with various nascent RNA folding strengths, inserted them into the yeast genome, and experimentally measured their nascent RNA structures with eSPET-seq. Three specific *CAN1* mutants with strong, intermediate and weak nascent RNA folding (Figure R7b) were selected and tested for the corresponding spontaneous mutation rate by the yeast *CAN1* forward mutation assay⁸. We found that the mutation rate is indeed lower for *CAN1* mutants with stronger nascent RNA folding (Figure R7c), suggesting that nascent RNA folding can attenuate spontaneous mutations.

To further verify that the reduction of mutation rate by nascent RNA folding is mediated by inhibiting R-loop formation, we examined the prevalence of R-loops at the 5' end of these three *CAN1* mutants by DRIP-RT-qPCR. Consistent with our model, we found that the R-loop signal was significantly reduced when there was strong nascent RNA folding (Figure R7d). To directly confirm that the antimutator effect of nascent RNA folding is R-loop-dependent, we stably overexpressed *RNASEH1* in the yeast genome, which hampers R-loop formation by degrading the RNA in an R-loop. It is predicted that *RNASEH1* overexpression reduces or even abolishes the effects of nascent RNA folding on spontaneous mutation rates, and this is indeed observed (compare Figure R7c and e). Specifically, the significant reduction in mutation rate of *CAN1* sequence with intermediate folding compared to that with weak folding is no longer significant upon *RNASEH1* overexpression. Similarly, the antimutator effect of strongly-folded *CAN1* relative to the other two *CAN1* sequences was also weakened, although it remained statistically significant. More importantly, such a disappearance/weakening of the antimutator effect of nascent RNA folding was accompanied by the disappearance/weakening of R-loop prevalence reduction upon *RNASEH1* overexpression (contrasting Figure R7d and f).

These observations made in the manipulative experiment of *CAN1* collectively demonstrate the R-loop-dependent antimutator effect of nascent RNA folding, and

add strong and specific support for our model. These results have now been added to our manuscript as Supplementary Fig. 8 and mentioned in Results.

Figure R7. Validating the R-loop-dependent antimutator effect of nascent RNA folding via manipulative experiments. (a) Snapshot of R-loop signals from S1-DRIP-seq. The y-axis represents the enrichment of S1-DRIP-seq reads relative to Input chromatin. There is a strong R-loop signal in the 1-300bp region. (b) Experimentally determined nascent RNA folding strengths of the three selected *CAN1* mutants with weak, intermediate and strong nascent RNA folding. (c) Relative mutation rate of *CAN1* for weak, intermediate and strong versions. (d) Relative R-loop signal at *CAN1* for weak, intermediate and strong versions. (e) Relative mutation rate at *CAN1* for weak, intermediate and strong versions when *RNASEH1* was overexpressed. (f) Relative R-loop signal at *CAN1* for weak, intermediate and strong versions when *RNASEH1* was overexpressed. All *P*-values are based on two-tailed *t* tests.

Reviewer #2's Comments

Overall comment

In this work, Yu and collaborators adapted the SPETseq method to study RNA cotranscriptional structure in *S. cerevisiae*. The new method, which the authors named eSPETseq for eukaryote SPETseq is improved with respect to the previous version by the use of NAI-N3-dependent modification of all four bases instead of DMS which could modify only A and C nucleotides, and enriched for the chromatin fraction. This improved method substantially confirmed in yeast the results previously shown in bacteria confirming that the folding is mainly due to intrinsic RNA properties. The authors go further proposing a mechanism by which the RNA cotranscriptional folding at the 5' end, competing with the R-loop, would be protective toward the mutagenesis originated on the single stranded DNA generated as a consequence of the R-loop.

Response:

We would like to thank the reviewer for his/her overall positive comments of our work.

Comment 1

The first part of the paper is solid and the technical improvement is substantial. However, it comes late as nowadays more refined methods have been developed for the analysis of RNA structures, which rely on mutations profiles and take also into account of the cellular RNA heterogeneity.

Response:

We thank the reviewer for his/her comments. We acknowledge that mutation profiling (MaP) methods, such as SHAPE-MaP or DMS-MaP are more refined methods for probing RNA structures. In spite of this, we do NOT believe the currently published MaP-based methods are superior to ours, particularly when it comes to

probing the local structure near the transcription site. We will briefly summarize our reasoning below, but in general, that indicates that eSPET-seq has a unique capability that cannot be replaced by other approaches.

First, eSPET-seq was designed with the aim of understanding the effect of nascent RNA folding on R-loops and spontaneous mutations in DNA. In order to fulfill this ultimate objective, it is imperative to obtain the position of the transcription site as well as to enrich the structural information near the transcription site, as this is likely to be the structural element that has the largest impact (if any) on the R-loop and mutation rate, given its proximity to the transcription site. The eSPET-seq protocol facilitates this by enriching the biotinylated SHAPE reactive site with Streptavidin beads, and enriching the transcription site with a ligation reaction that targets the 3' hydroxyl terminus of the nascent transcript. Compared to MaP-based methods (which cannot enrich, i.e., SHAPE-reactive sites), these designs greatly enhance the number of informative reads in eSPET-seq.

Second, MaP is not without its own disadvantages. In the case of SHAPE-MaP, for example, a substantial number of indels will be generated (~40%). Many deletions are difficult to locate unambiguously with single-nucleotide accuracy, particularly in regions of repetitive sequences. In typical RNAs, 55% of the detected deletions (22% of total mutations) are aligned ambiguously, which is usually excluded from further analysis in order to produce more accurate secondary structure models¹⁰. Although DMS-MaP has lower indel rates (~6%) than SHAPE-MaP, it is limited to detecting structural information about A and C bases¹¹. For molecules with high abundance, such as mature RNAs, these limitations could be fine. But for RNAs with low abundance, such as transcriptional intermediates, they could be highly problematic, not to mention that we are only targeting the 100 nt region closest to the transcription site.

Third, from a more quantitative angle, in SHAPE-MaP, only about 2 out of 100 nucleotides are modified. Due to the fact that the reverse transcriptase cannot 100% misread a SHAPE-modified nucleotide, only ~50% of the chemical adducts are

detected¹⁰, which equates to ~1 modification per 100 bases. In this sense, it is effectively equivalent to the single-hit kinetics (one modification per 100 bases) of eSPET-seq. In this regard, eSPET-seq should provide a similar resolution for the structure compared with SHAPE-MaP for the 100 nt region near the transcription site.

Finally, for the cellular RNA heterogeneity. We fully appreciate the importance of structural heterogeneity. In fact, we have published a paper on this topic¹². Nevertheless, such heterogeneity has two facets. In the case of mature RNAs, structural heterogeneity between transcripts of the same gene is essential since there may be a large number of transcripts derived from the same gene that have the same sequence but differ in structure. When it comes to nascent RNA, however, heterogeneity among transcriptional intermediates of different lengths is clearly more significant. This is because within a diploid cell, there could only be at most two transcriptional intermediates from the same gene with exactly the same length (since there are only two template DNAs). On the contrary, there could be many more different intermediates of different lengths. More importantly, as RNA molecules are transcribed, their structures will change as transcription proceeds (i.e. co-transcriptional folding)¹³. The eSPET-seq has unique strength in resolving the latter type of heterogeneity, which is more important for nascent RNA and is therefore more relevant to the main topic of our study.

Together, eSPET-seq provides a unique opportunity to study how nascent RNA folding affects R-loops and spontaneous mutations of DNA. We hope that our reasoning will convince the reviewer of the necessity of eSPET-seq.

Comment 2

The second part of the work, namely the model by which RNA structure would be protective on DNA mutagenesis, is mainly speculative. More solid data on specific models should be performed to substantiate the proposed model.

Response:

Thank you again for this very constructive suggestion, which was also made by Reviewer#1 (Comment 6). In order to further confirm our findings, we have now conducted additional manipulative experiments and analyses, which is briefly summarized below.

Our manipulative experiment focused on the *CANI* gene, which encodes an arginine permease that is commonly used to detect spontaneous mutations. It has been demonstrated⁸ that a functional *CANI* is lethal in media containing the toxic arginine analog canavanine, whereas a *CANI* null mutant is viable. By measuring the frequency of colonies that were resistant to canavanine, the spontaneous mutation rate in the *CANI* gene can be estimated. Furthermore, S1-DRIP-seq data demonstrated that *CANI* forms R-loops in the first 300 bp of its 5' end during transcription (Figure R8a), thereby enabling manipulative experiment.

Utilizing this reporter gene, we first determined whether increased nascent RNA folding would reduce spontaneous mutations. To manipulate nascent RNA folding strength around the region prone to form R-loop (the first 300 bp of *CANI*'s 5' end), we first generated synonymous mutants of *CANI* by shuffling all synonymous codons within this region, so that the protein sequence and the codon usage bias of these mutants remained identical to wild-type *CANI*. We then predicted the nascent RNA folding strength of these *CANI* mutants using DrTransformer⁹. Considering the potential error of *in silico* structure prediction, we synthesized several mutants with various nascent RNA folding strengths, inserted them into the yeast genome, and experimentally measured their nascent RNA structures with eSPET-seq. Three specific *CANI* mutants with strong, intermediate and weak nascent RNA folding (Figure R8b) were selected and tested for the corresponding spontaneous mutation rate by the yeast *CANI* forward mutation assay⁸. We found that the mutation rate is indeed lower for *CANI* mutants with stronger nascent RNA folding (Figure R8c), suggesting that nascent RNA folding can attenuate spontaneous mutations.

To further verify that the reduction of mutation rate by nascent RNA folding is mediated by inhibiting R-loop formation, we examined the prevalence of R-loops at the 5' end of these three *CANI* mutants by DRIP-RT-qPCR. Consistent with our model, we found that the R-loop signal was significantly reduced when there was strong nascent RNA folding (Figure R8d). To directly confirm that the antimutator effect of nascent RNA folding is R-loop-dependent, we stably overexpressed *RNASEHI* in the yeast genome, which hampers R-loop formation by degrading the RNA in an R-loop. It is predicted that *RNASEHI* overexpression reduces or even abolishes the effects of nascent RNA folding on spontaneous mutation rates, and this is indeed observed (compare Figure R8c and e). Specifically, the significant reduction in mutation rate of *CANI* sequence with intermediate folding compared to that with weak folding is no longer significant upon *RNASEHI* overexpression. Similarly, the antimutator effect of strongly-folded *CANI* relative to the other two *CANI* sequences was also weakened, although it remained statistically significant. More importantly, such a disappearance/weakening of the antimutator effect of nascent RNA folding was accompanied by the disappearance/weakening of R-loop prevalence reduction upon *RNASEHI* overexpression (contrasting Figure R8d and f).

These observations made in the manipulative experiment of *CANI* collectively demonstrate the R-loop-dependent antimutator effect of nascent RNA folding, and add strong and specific support for our model. These results have now been added to our manuscript as Supplementary Fig. 8 and mentioned in Results.

Figure R8. Validating the R-loop-dependent antimutator effect of nascent RNA folding via manipulative experiments. (a) Snapshot of R-loop signals from S1-DRIP-seq. The y-axis represents the enrichment of S1-DRIP-seq reads relative to Input chromatin. There is a strong R-loop signal in the 1-300bp region. (b) Experimentally determined nascent RNA folding strengths of the three selected *CAN1* mutants with weak, intermediate and strong nascent RNA folding. (c) Relative mutation rate of *CAN1* for weak, intermediate and strong versions. (d) Relative R-loop signal at *CAN1* for weak, intermediate and strong versions. (e) Relative mutation rate at *CAN1* for weak, intermediate and strong versions when *RNASEH1* was overexpressed. (f) Relative R-loop signal at *CAN1* for weak, intermediate and strong versions when *RNASEH1* was overexpressed. All *P*-values are based on two-tailed *t* tests.

Reviewer #3's Comments

Overall Comment

This paper has the dual function of examining the incidence and dynamics of mRNA folding in yeast, and of relating this to information on mutation rates.

Response:

We thank the reviewer for his/her supportive comment.

Comment 1

The method that they introduce to quantify / localize RNA folding does seem to be an advance over previous methods. However, Figure 1 and the description of the method needs to be substantially simplified / clarified. I am not entirely up on this kind of methodology, but I'm also not entirely ignorant, but I found the description of how one goes from starting raw RNA to estimated positions of stems and loops quite impenetrable. If other reviewers more in the know on the methodology are happy with this aspect of the paper, then it alone is a plus.

Response:

We apologize for not describing the method more clearly. We have now revised the Results to enhance simplicity and clarity. The revised texts are quoted below.

To this end, we adapted the Structural Probing of Elongating Transcripts (SPET-seq) method previously developed in prokaryotes¹⁴ to eukaryotes with several major improvements (Fig. 1a. See Methods for detailed experimental procedures). First, we used NAI-N₃, a chemical that is capable of modifying all four nucleotides (adenine (A), U, G, and cytosine (C))¹⁵, to probe single-stranded bases in nascent RNA (Fig. 1a). The full coverage for all nucleotides is an apparent advantage over DMS previously used in SPET-seq¹⁴, which can only modify A and C nucleotides. Second, we enriched the nascent RNA by using the chromatin fraction after cellular fractionation (see also Supplementary Fig. 1). Third, we ligated the 3' hydroxyl terminus of the nascent transcript with a 3' adapter, which was later paired with a primer for reverse transcription (RT). Nascent RNA segments near the transcription site were enriched during this step due to the presence of terminal phosphates in hydrolysis and degradation products. Next, after RT that should stop at the NAI-N₃-modified

nucleotide and enrichment for biotinylated molecules, the cDNA was extracted and ligated with a 5' adapter. Finally, the adapter-linked cDNA was PCR-amplified before being subjected to paired-end HTS (Fig. 1a), in which the forward reads represented the unpaired nucleotides tagged by NAI-N₃, and the reverse reads represented the transcription site. As a control for the RT stops triggered by factors other than NAI-N₃ modification, we also performed parallel experiments without NAI-N₃ treatment, in which biotinylated 3' adapters were used. This experimental procedure (named eSPET-seq, "e" for eukaryote) collectively enabled localization of single-stranded bases (those at NAI-N₃-dependent RT stops), which can be further utilized to approximate the secondary structures of nascent RNA.

In addition, we have deposited the source codes to GitHub (https://github.com/GongwangYu/NascentRNAfolding_NS-seq) for the entire computational pipeline from the raw sequencing data to the single-stranded scores, which will further enhance the interpretability and repeatability of eSPET-seq. We hope these would make our methods more accessible to the reviewer and prospective readers.

Comment 2

Bigger issues arise with respect to the analyses of mutational vulnerability, and I'm not convinced this should be included.

In particular, on line 220 – measuring mutation rates with DNA polymorphism data does not appear to be trustworthy. This not only reflects mutation, but also selection on individual sites, and the latter through both direct selection on sites and through local reductions in effective population sizes owing to selection on linked sites. Not only do the authors ignore the functional significance of the site, which will influence the standing levels of variation in potentially very dramatic ways (assuming there is indeed selection on folding), but they set mutation rates equal to zero when no polymorphisms are observed (which is a significant issue with sampling error associated with only moderate numbers of sequenced genomes).

So, overall, I am not confident in this part of the analysis. To their credit, the authors conclude that the strength of selection is insufficient for selection to be paying a role in generating the supposed local variation in mutation rates, which was a relief to see. I'll note in addition that prior work with mutation-accumulation experiments in a diversity of species has failed to find a compelling link between transcription and mutation rates, so this needs to be cited as well.

Response:

The reviewer's comments are greatly appreciated. We would like to respond from two different perspectives.

Firstly, we agree that using DNA polymorphisms to estimate spontaneous mutation rates is not ideal, because some polymorphisms may be subject to natural selection. Nevertheless, it should be noted that the use of polymorphism-based mutation rate estimation is more of a necessary compromise in terms of feasibility. On the one hand, other alternatives (such as mutation accumulation experiments, as mentioned by the reviewer) have a more serious issue of scarcity of mutations, making the subsequent quantification of spontaneous mutation rates for a large number of single-bases within the genome more difficult, and the subsequent statistical tests less effective (compared with polymorphisms). On the other hand, it is generally believed that the vast majority of DNA polymorphisms are neutral, with only a few being affected by natural selection¹⁶. As a result, under certain conditions, such as when single-base mutation rates are required, DNA polymorphisms are widely used as a reasonable approximation¹⁷⁻²¹.

Second, in an effort to address the reviewer's comment in a more quantitative manner, we compared the mutation rates we estimated based on the DNA polymorphism among the 190 yeast strains with those estimated by mutation accumulation experiments^{22,23}. We found that these two metrics are highly correlated (**Figure R9**, also included as Supplementary Fig. 11 in the revised manuscript), and they both support the antimutator effect of nascent RNA folding (**Figure R10**. See below in the response to the next comment, also included as Supplementary Fig. 7 in the revised manuscript).

Figure R9. Correlation between the spontaneous mutation rate estimates derived from DNA polymorphisms of 190 *S. cerevisiae* strains and mutation accumulation experiments. The yeast genes with polymorphism-based mutation rate estimates are divided into 10 equal-sized groups with increasing mutation rate (x axis). The average mutation rate of each group as estimated from the mutation accumulation experiments was calculated (y axis) and found to be positively correlated with the polymorphism-based mutation rate. Error bars indicate the 95% confidence interval of the mean, estimated by bootstrapping the genes 1,000 times. Spearman's rank correlation and its significance (P value) is shown.

Comment 3

Although these authors emphasize tam, transcription-associated mutagenesis, there is also a phenomenon called transcription-coupled repair, so it is unclear that the former effect dominates. They do seem to try to check this, although one of the references they cite does not seem to be a mutation-accumulation experiment, so the overall test seems not to be strong (and they do not discuss statistical significance). Here, there is also the issue of the rate of elongation of individual transcripts vs. the rate of overall transcript production (which is presumably driven by initiation events). MA experiments simply do not generate enough data to perform fine-scaled analyses, but perhaps the authors could do a simple test to whether binned regions with and without secondary structure have different mutation rates, factoring out the problem of expression level / rate.

Response:

We would like to thank the reviewer for his/her comments and advice. As for the relative effect size of TCR and TAM, several genome-wide studies in bacteria, budding yeast and the human germline have demonstrated that spontaneous mutation rates tend to increase with gene expression levels, which is consistent with the notion that TAM has a greater impact than TCR²⁴⁻²⁷. We have also re-examined the two references cited in this context, namely^{22,23}. We have confirmed that the datasets we used from these two references are indeed both derived from mutational accumulation experiments. We have now further extended this section related to TCR to better clarify our logic, which is pasted below.

We inferred with eSPET-seq data that the increased TAM in highly expressed genes was weakened by the stronger nascent RNA folding (Fig. 5a and b). This observation could have been confounded by transcription-coupled repair (TCR), as TCR is theoretically more active for highly expressed genes²⁸ (but see ²⁹). To disentangle the antimutator effect of nascent RNA folding and that of TCR, we exploited the different strand biases of the two mechanisms. Specifically, TCR repairs mutations on the template strand³⁰ and nascent RNA folding suppresses mutations on the coding strand⁸. We collected spontaneous mutation events found in two mutation accumulation experiments^{22,23} and classified all C/G to T/A mutations into either template or coding strand by assuming dominant contribution by hydrolytic deamination of cytosine in C/G to T/A mutations³¹ (See Materials and Methods). Based on these stranded mutations, we found that the prevalence of nascent RNA structure was anticorrelated with the relative C-to-T mutation rate of the coding strand (Supplementary Figure 8a. Spearman's $\rho = -0.28$, $P < 0.05$) but not that of the template strand (Supplementary Figure 8b. Spearman's $\rho = -0.04$, $P = 0.75$), thereby suggesting the TCR-independent antimutator effect of nascent RNA folding on the coding strand. In terms of effect size, comparing the top and bottom 20% of genes in terms of prevalence of nascent RNA structure revealed a 25% decrease of C-to-T mutation rate of the coding strand. We further contrast the C-to-T mutation rate of template and coding strand to better isolate the TCR-independent antimutator effect of nascent RNA structure (Supplementary Figure 8c and d). Specifically, we found that C-to-T mutation rate of template strand (suppressed by TCR) relative to that of the coding strand (suppressed by nascent RNA folding) decreased 21% for highly expressed genes compared to lowly expressed genes (Supplementary Figure S8c), which revealed the effect of TCR; And the C-to-T mutation rate of coding strand (suppressed by nascent RNA folding) relative to that of the template strand (suppressed by TCR) decrease by 24% for genes with high prevalence of nascent RNA structure compared to genes with low prevalence of nascent RNA structure (Supplementary Figure S8d), which revealed

the antimutator effect of nascent RNA folding. Collectively, these results suggested substantial antimutator effect by nascent RNA folding independent of TCR.

Per the reviewer's suggestion regarding the mutation accumulation experiments, we have also compared the prevalence of nascent RNA structure in regions with and without mutations (binning by mutated or not is preferred over binning by secondary structure due to the scarcity of mutations in MA datasets). We found that non-mutated sites have significantly stronger RNA folding compared to mutated sites (**Figure R10** below). This result has also been added as Supplementary Fig. 7 in the revised manuscript.

Figure. R10. Mutated sites in mutation accumulation experiments^{22,23} tend to have lower prevalence of nascent RNA structure compared to non-mutated sites in the same gene. Violin plot for prevalence of nascent RNA secondary structure in mutated versus non-mutated sites in mutational accumulation experiments. The prevalence of nascent RNA secondary structure was approximated by either negative value (a) or Gini index (b) of single-stranded scores. The black dot inside the violin shows the mean. *P*-values are based on Wilcoxon signed-rank tests.

Comment 4

An explanation / definition is needed for the Gini index. Generally, this index is used as a measure of uniformity, and it just wasn't clear what the authors were doing here.

Response:

Thanks a lot for this question. It has previously been shown that nucleotides within RNA segments with complex secondary structures should display a high degree of inequality in single-stranded scores^{24,25}. Accordingly, these previous studies approximated the prevalence of secondary structures within the focal region using the Gini index of single-stranded scores. We followed their practice, and further backed up the conclusion with results derived from the negative value of the single-stranded score. Our manuscript has now been revised to clarify this usage of Gini index. The related text in Result is pasted below.

Here the single-stranded score was first calculated for each nucleotide (see Methods), reflecting the relative probability of it being unpaired/single-stranded, and then aggregated for a per-gene metric of prevalence of nascent RNA structure by the negated mean within the gene or by Gini index (a higher Gini index indicates a more structured region³²) (Supplementary Fig. 5a and b).

And the related text in Method is pasted below.

We also calculated the Gini index (by R package "ineq") using the single-stranded scores of all nucleotides within a gene to represent the average prevalence of RNA secondary structure of the gene, as it has previously been shown that as the structure unfolds, the single-stranded score becomes more even (low Gini index)³².

Comment 5

Finally, there are many grammatical errors that I have not attempted to correct, and this will need to be fixed prior to publication.

Response:

Thank you for pointing out the grammatical errors. We apologize for the errors. Our revised manuscript has been carefully reviewed and we have made every effort to

improve clarity and avoid any other typographical errors. Detailed modifications are highlighted in the revised manuscript.

References

1. Subtelny, A.O., Eichhorn, S.W., Chen, G.R., Sive, H. & Bartel, D.P. Poly(A)-tail profiling reveals an embryonic switch in translational control. *Nature* **508**, 66-71 (2014).
2. Oesterreich, F.C., *et al.* Splicing of Nascent RNA Coincides with Intron Exit from RNA Polymerase II. *Cell* **165**, 372-381 (2016).
3. Ding, Y., *et al.* In vivo genome-wide profiling of RNA secondary structure reveals novel regulatory features. *Nature* **505**, 696-700 (2014).
4. Marinus, T., Fessler, A.B., Ogle, C.A. & Incarnato, D. A novel SHAPE reagent enables the analysis of RNA structure in living cells with unprecedented accuracy. *Nucleic Acids Res* **49**, e34 (2021).
5. Giannetti, C.A., Busan, S., Weidmann, C.A. & Weeks, K.M. SHAPE Probing Reveals Human rRNAs Are Largely Unfolded in Solution. *Biochemistry* **58**, 3377-3385 (2019).
6. Simon, L.M., *et al.* In vivo analysis of influenza A mRNA secondary structures identifies critical regulatory motifs. *Nucleic Acids Res* **47**, 7003-7017 (2019).
7. Deigan, K.E., Li, T.W., Mathews, D.H. & Weeks, K.M. Accurate SHAPE-directed RNA structure determination. *Proc Natl Acad Sci U S A* **106**, 97-102 (2009).
8. Chen, X., Yang, J.-R. & Zhang, J. Nascent RNA folding mitigates transcription-associated mutagenesis. *Genome Research* **26**, 50-59 (2016).
9. Badelt, S., Lorenz, R. & Hofacker, I.L. DrTransformer: heuristic cotranscriptional RNA folding using the nearest neighbor energy model. *Bioinformatics* **39**(2023).
10. Smola, M.J., Rice, G.M., Busan, S., Siegfried, N.A. & Weeks, K.M. Selective 2'-hydroxyl acylation analyzed by primer extension and mutational profiling (SHAPE-MaP) for direct, versatile and accurate RNA structure analysis. *Nat Protoc* **10**, 1643-1669 (2015).
11. Zubradt, M., *et al.* DMS-MaPseq for genome-wide or targeted RNA structure probing in vivo. *Nat Methods* **14**, 75-82 (2017).
12. Yu, G., Zhu, H., Chen, X. & Yang, J.R. Specificity of mRNA Folding and Its Association with Evolutionarily Adaptive mRNA Secondary Structures. *Genomics Proteomics Bioinformatics* **19**, 882-900 (2021).
13. Spitale, R.C. & Incarnato, D. Probing the dynamic RNA structurome and its functions. *Nat Rev Genet* **24**, 178-196 (2023).

14. Incarnato, D., *et al.* In vivo probing of nascent RNA structures reveals principles of cotranscriptional folding. *Nucleic Acids Res* **45**, 9716-9725 (2017).
15. Flynn, R.A., *et al.* Transcriptome-wide interrogation of RNA secondary structure in living cells with icSHAPE. *Nature Protocols* **11**, 273-290 (2016).
16. Bowcock, A.M., *et al.* Drift, admixture, and selection in human evolution: a study with DNA polymorphisms. *Proceedings of the National Academy of Sciences of the United States of America* **88**, 839-843 (1991).
17. Liu, X., Maxwell, T.J., Boerwinkle, E. & Fu, Y.X. Inferring population mutation rate and sequencing error rate using the SNP frequency spectrum in a sample of DNA sequences. *Molecular biology and evolution* **26**, 1479-1490 (2009).
18. Lercher, M.J. & Hurst, L.D. Human SNP variability and mutation rate are higher in regions of high recombination. *Trends Genet* **18**, 337-340 (2002).
19. Fu, Y.X. Estimating mutation rate and generation time from longitudinal samples of DNA sequences. *Molecular biology and evolution* **18**, 620-626 (2001).
20. Klein, E.K., Austerlitz, F. & Laredo, C. Some statistical improvements for estimating population size and mutation rate from segregating sites in DNA sequences. *Theor Popul Biol* **55**, 235-247 (1999).
21. Fu, Y.X. Estimating effective population size or mutation rate using the frequencies of mutations of various classes in a sample of DNA sequences. *Genetics* **138**, 1375-1386 (1994).
22. Fares, M.A., Keane, O.M., Toft, C., Carretero-Paulet, L. & Jones, G.W. The roles of whole-genome and small-scale duplications in the functional specialization of *Saccharomyces cerevisiae* genes. *PLoS Genet* **9**, e1003176 (2013).
23. Zhu, Y.O., Siegal, M.L., Hall, D.W. & Petrov, D.A. Precise estimates of mutation rate and spectrum in yeast. *Proceedings of the National Academy of Sciences of the United States of America* **111**, E2310-2318 (2014).
24. Lind, P.A. & Andersson, D.I. Whole-genome mutational biases in bacteria. *Proc Natl Acad Sci U S A* **105**, 17878-17883 (2008).
25. Chen, X. & Zhang, J. Yeast mutation accumulation experiment supports elevated mutation rates at highly transcribed sites. *Proc Natl Acad Sci U S A* **111**, E4062 (2014).
26. Chen, X. & Zhang, J. No gene-specific optimization of mutation rate in *Escherichia coli*. *Mol Biol Evol* **30**, 1559-1562 (2013).
27. Park, C., Qian, W. & Zhang, J. Genomic evidence for elevated mutation rates in highly expressed genes. *EMBO Rep* **13**, 1123-1129 (2012).
28. Mellon, I., Bohr, V.A., Smith, C.A. & Hanawalt, P.C. Preferential DNA repair of an active gene in human cells. *Proc Natl Acad Sci U S A* **83**, 8878-8882 (1986).
29. Keightley, P.D., *et al.* Analysis of the genome sequences of three *Drosophila melanogaster* spontaneous mutation accumulation lines. *Genome Res* **19**,

- 1195-1201 (2009).
30. van den Heuvel, D., van der Weegen, Y., Boer, D.E.C., Ogi, T. & Lijsterburg, M.S. Transcription-Coupled DNA Repair: From Mechanism to Human Disorder. *Trends in cell biology* **31**, 359-371 (2021).
 31. Maki, H. Origins of spontaneous mutations: specificity and directionality of base-substitution, frameshift, and sequence-substitution mutageneses. *Annu Rev Genet* **36**, 279-303 (2002).
 32. Rouskin, S., Zubradt, M., Washietl, S., Kellis, M. & Weissman, J.S. Genome-wide probing of RNA structure reveals active unfolding of mRNA structures in vivo. *Nature* **505**, 701-705 (2014).

Reviewers' comments:

Reviewer #1 (Remarks to the Author):

I appreciate that the authors have spent a lot of time addressing the reviewer's concerns. However, I still have some concerns with regards to data quality-

- 1) In Figure R2- the plot of number of genes/read depth is pretty much linear- that means that the authors are way below the saturation limit. I am worried that the threshold of 1 average RT stop/nt might be too low- if the authors had used a more stringent cutoff, do the results hold?
- 2) With regards to Figure R3b- the authors show that the structural switch occurs after base 80, could the authors RT using a primer between base 80-120bases?

Reviewer #3 (Remarks to the Author):

Along with introducing some potentially novel results on rna folding using a new method, the authors primarily weave their narrative around the claim that such structure reduces local mutation rates in DNA. Unfortunately, the arguments here on causality are not very convincing, and seemingly less so after this revision. One gets the point that the authors are valiantly trying to prove a particular point rather than taking a fully objective view of the data. As I cannot provide good input into the molecular methodologies introduced herein, I will confine my comments to the analyses of mutation rates.

1) To gain information on site-specific mutation rates, the authors apparently use information on natural variation in yeast. However, their argument that polymorphism provides a good estimate of mutation rates is not convincing, especially in yeast, and the references they cite do not support this and mostly are not even relevant to the topic. Figure R9 is not terribly convincing, and it would be helpful to know how the MA-based estimates were obtained and compiled. As pointed out in my prior review, the problem with polymorphism-based mutation-rate inference is the operation of selection on mutations in nature. Perhaps a little more confidence would arise here if the authors confined their analyses to silent-sites, but even here there may be problems if the recent claim of strong selection on silent sites in yeast is correct.

2) All this being said, the authors don't actually ever define their measure of mutation (as far as I can tell, even in the methods section). They call it a polymorphism-based estimate, but in Figures 4d-f the approximate means appear to be on the order of 0.20. This does make sense. It seems far too high to be the conventional measure of nucleotide diversity (virtual heterozygosity), and probably too high for a simple measure of the fraction of sites exhibiting polymorphism (which would not be very informative, as essentially every site in yeast is probably polymorphic on a global basis). Even if it is based on polymorphism in some way, I thought there were more on the order of 1000 yeast strains sequenced now, not 109.

3) Figures 4d-f are also far from convincing. The authors give very low P values for the regressions, but they never say how the regressions were performed. For example, are they polynomial regressions, and if so, how many terms were included, and what was the statistical justification for adding terms? Why are the regressions truncated within the range of the data? One can see that for at least 50% of the range of data, the regressions are effectively flat, i.e., there is no response of the mutation rate to the dependent variable, and even if one takes into consideration the overall fits (which may be overdetermined), the model only explains 4% of the variation in the data.

4) On lines 250-252, the authors note regression coefficients of -0.18 and -0.09, and then conclude that 35% and 65% of the variance is explained by the model. I'm afraid that I have no idea as to how such a conclusion has been reached. The fraction of variance explained by a model is equal to the square of the correlation coefficient, 3.2 and 0.8% for the numbers given above. Perhaps I'm missing

something here, but throughout the paper one can see by eye alone that the patterns being advocated are simply not very strong.

5) It is also difficult to know how much faith to put into the Can1 reporter-construct results referred to from the supplemental table 8. The problem here is that reporter-construct mutation-rate estimates are known to be only weakly correlated with overall genomic mutation rates, and here we have the potential additional problem that the constructs produced by the authors may have effects on the mutation rate completely independent of rna structural effects.

6) On lines 305-308, the authors suggest that they compared the evolutionary conservation of sequences to structural features. However, although they suggest that the approach used to estimate the degree of conservation is in the Methods section, I could not find it. They may be referring to the polymorphism data, but it is well known that polymorphisms are enriched for variants that never go to fixation, so that such indices are not appropriate measures of functional constraint. A more appropriate approach is to use between-species divergence data, ideally normalize by polymorphism data (e.g., as a neutrality index).

7) Finally, the cancer data outlined in Figures 6a-f are even more unconvincing in terms of identifying a mutational effect. Again, the authors fit a model with apparently at least three (based on the inflections in the curvilinearity (and perhaps more) coefficients, with the overall fits being extremely flat.

Unfortunately, my review of the revision is more negative than the review of the original submission, mainly as I had hoped that some of the above issues would have been dealt with in a convincing way. Possibly, a reviewer more focused on the estimation of rna folding estimation would be more favorably inclined, but I do not find the linking of such information to mutational susceptibility to be convincing.

There remain many grammatical errors and typos in the paper.

Reviewer #1 (Remarks to the Author):

Overall comments

I appreciate that the authors have spent a lot of time addressing the reviewer's concerns. However, I still have some concerns with regards to data quality-

Response:

We would like to thank the reviewer for his/her constructive suggestions. Please see below our responses regarding the data quality.

Comment 1

In Figure R2- the plot of number of genes/read depth is pretty much linear- that means that the authors are way below the saturation limit. I am worried that the threshold of 1 average RT stop/nt might be too low- if the authors had used a more stringent cutoff, do the results hold?

Response:

Thank you very much. Due to the fact that up-sampling (the dots to the right of the red dot, representing simulated/up-sampled data with higher sequencing coverage than the actual data) is only carried out by computational simulation, no saturation is expected as occurred in the real experiment. As an example, imagine a gene that is rarely captured regardless of how deep the sequencing is. This could be a turned-off gene with some leaky expression, or just rarely captured due to technical issues/sequencing biases (see below). As is common in HTS experiments with saturating depth, increasing the sequencing depth will not result in a proportional increase in the coverage of this gene. However, as long as one read is (by chance) captured for this gene, its coverage will increase proportionally to the sequencing depth in the simulated up-sampling, since the sequencing bias against this gene in real HTS experiments does not exist in the simulated up-sampling. Several factors may contribute to these biases, including, but not limited to, a biased nucleotide composition hindering accurate sequencing, or a scarcity of starting template molecules that cannot be resolved satisfactorily by PCR amplification prior to sequencing. These issues are not modeled by a simple simulation of up-sampling.

Per the reviewer's suggestion, we've tried more stringent cutoffs. All results hold, as shown below. Unless otherwise stated, the results are listed for each relevant result figure panel, with the

left-most result being the one from the main figure using the threshold of **1 stop/nt**, and the **middle** and **right-most** results from the thresholds of **2 stop/nt** and **4 top/nt**, respectively.

Fig.3a. Structural similarity between nascent and mature RNA. Most genes displayed significant similarity, although the number of genes reduced due to increased threshold.

Fig.3d. The structural similarity between mature RNAs folded *in vivo* and *in vitro* was compared to that between nascent and mature RNAs folded *in vivo*. Regardless of the threshold used, the results showed a consistent positive correlation.

Fig.3e. Standard boxplot showing that the top 50% of genes with higher structural similarity between mature RNAs folded *in vivo* and *in vitro* tended to have less RBP binding sites than the other genes. Regardless of the threshold used, the results remained significant by Mann-Whitney test.

Fig.3f. Standard boxplots showing that the top 50% of genes with higher structural similarity between nascent and mature RNAs folded in vivo tended to have less RBP binding sites than the other genes. Regardless of the threshold used, the results remained significant by Mann-Whitney test.

Fig.4b. The negative of single-stranded score was compared to the R-loop score of the same gene. Regardless of the threshold used, the results showed a consistent negative correlation.

Fig.4c. The Gini index of single-stranded score was compared to the R-loop score of the same gene. Regardless of the threshold used, the results showed a consistent negative correlation.

Fig.4e. The negative of single-stranded score was compared to the mutation rate of the same gene. Regardless of the threshold used, the results showed a consistent negative correlation.

Fig.4f. The Gini index of single-stranded score was compared to the mutation rate of the same gene. Regardless of the threshold used, the results showed a consistent negative correlation.

Fig.4g. The R-loop-dependent fraction of antimutator effect by nascent RNA folding was estimated by contrasting the correlation and the partial correlation (with R-loop score controlled) between the mutation rate and the prevalence of nascent RNA structure. The fraction of R-loop dependent antimutagenic effects by negative of and Gini index of single-stranded score ranged from 35% to 54% and 68% to 98% ,respectively, across the three threshold conditions. These results consistently supported our conclusion that a large fraction of the antimutator effect of nascent folding was mediated via the R-loop.

Fig.4i. Three odds ratios representing the correspondence among single-stranded scores, R-loop scores and mutation rates (OR_{S-R} , OR_{R-M} and OR_{S-M}) were calculated for each gene, and then combined and tested for statistical significance by the Cochran-Mantel-Haenszel chi-squared test. Regardless of the threshold used, the results significantly supported our hypothesized model of nascent RNA folding mitigating TAM by dissolving the R-loop.

Fig.4j. Three within-gene correlations among negative of single-stranded score, R-loop score and mutation rate of nucleotide positions were averaged (red arrow) and compared with their random expectations (histogram), which were estimated by permutating the negative of single-stranded score, R-loop score and mutation rate within each gene 1,000 times. P values from permutation tests are indicated. The three panels at the top represent results from 1 stop/nt, while those at the middle represent results from 2 stops/nt, and those at the bottom represent results from 4 stops/nt. Regardless of the threshold used, the results significantly supported our hypothesized model of nascent RNA folding mitigating TAM by dissolving the R-loop.

Fig. 5a. The expression level of a gene was compared to the negative of single-stranded score of the same gene. Regardless of the threshold used, the results showed a consistent positive correlation.

Fig. 5b. The expression level of a gene was compared to the Gini index of single-stranded score of the same gene. Regardless of the threshold used, the results showed a consistent positive correlation.

Fig. 5c. The evolutionary conservation of a gene was compared to the negative of single-stranded score of the same gene. Regardless of the threshold used, the results showed a consistent positive correlation.

Fig. 5d. The evolutionary conservation of a gene was compared to the Gini index of single-stranded score of the same gene. Regardless of the threshold used, the results showed a consistent positive correlation.

In summary, all the results and conclusions are robust to more stringent cutoffs.

Comment 2

With regards to Figure R3b- the authors show that the structural switch occurs after base 80, could the authors RT using a primer between base 80-120bases?

Response:

Thanks for your suggestion. In fact, the downstream primer L2 is reverse complementary to nucleotides 95-115 of the 5S RNA. We apologize for not making this clearer previously in Figure R3 and Figure S4.

In response to the reviewer's comment, and to improve the resolution of the co-transcriptional change in the R-loop signal, we have added an additional RT primer (nucleotides 90-105) between L1 (nucleotides 58-78) and L2 (nucleotides 95-115). We have also revised the related figure to clearly indicate the position of the various RT-primers. In addition to being pasted below as Figure RR3, this revised figure is also added as Supplementary Figure 4.

Most importantly, the R-loop dissolution after transcription passed 80 nucleotides was still evident, and the pattern still disappeared upon overexpression of *RNASEH1*. Our previous conclusion has been confirmed with enhanced resolution by this result.

Reviewer #3 (Remarks to the Author):

Overall comments

Along with introducing some potentially novel results on rna folding using a new method, the authors primarily weave their narrative around the claim that such structure reduces local mutation rates in DNA. Unfortunately, the arguments here on causality are not very convincing, and seemingly less so after this revision. One gets the point that the authors are valiantly trying to prove a particular point rather than taking a fully objective view of the data. As I cannot provide good input into the molecular methodologies introduced herein, I will confine my comments to the analyses of mutation rates.

Response:

Thank you for taking the time to review our manuscript. We have indeed focused our study over the specific hypothesis of the antimutator effect of nascent RNA structure. It is our hope that the revisions and clarifications listed below will resolve the problems to your satisfaction.

Comment 1 & 2

To gain information on site-specific mutation rates, the authors apparently use information on natural variation in yeast. However, their argument that polymorphism provides a good estimate of mutation rates is not convincing, especially in yeast, and the references they cite do not support this and mostly are not even relevant to the topic. Figure R9 is not terribly convincing, and it would be helpful to know how the MA-based estimates were obtained and compiled. As pointed out in my prior review, the problem with polymorphism-based mutation-rate inference is the operation of selection on mutations in nature. Perhaps a little more confidence would arise here if the authors confined their analyses to silent-sites, but even here there may be problems if the recent claim of strong selection on silent sites in yeast is correct.

All this being said, the authors don't actually ever define their measure of mutation (as far as I can tell, even in the methods section). They call it a polymorphism-based estimate, but in Figures 4d-f the approximate means appear to be on the order of 0.20. This does make sense. It seems far too high to be the conventional measure of nucleotide diversity (virtual heterozygosity), and probably too high for a simple measure of the fraction of sites exhibiting polymorphism (which would not be very informative, as essentially every site in yeast is probably polymorphic on a global basis). Even if it is based on polymorphism in some way, I thought there were more on the order of 1000 yeast strains sequenced now, not 109.

Response:

These two comments are both related to the polymorphism-based estimation of mutation rate, so we will answer them together. **First**, regarding the exact nature of the polymorphism-based estimate of mutation rate. We have briefly mentioned it as calculated by the GAMMA algorithm ¹, which is widely used and has been cited for 159 times according to Google Scholar. We understand that this brief description has not been sufficiently clear for the reviewer, so we have extended this Method section to explain more details underlying the GAMMA algorithm. This extended text is pasted below.

All single nucleotide variations (SNVs) from previously compiled population genomic data of 190 *S. cerevisiae* strains, as well as their inferred phylogenetic relationship, were extracted from the original publication². Based on this dataset, we applied the GAMMA algorithm ¹ to estimate the (relative) mutation rate for each segregating site, as described below. First, based on the phylogeny and the genotypes of the focal site, GAMMA first inferred its ancestral state on each internal node as the one with the highest (posterior) probability using a distance-based method ³. Second, GAMMA estimates the expected number of mutations per unit of time at the focal site using a maximum likelihood (ML) method, which takes into account the possibility of multiple mutations appearing as only one nucleotide change in a long branch. Specifically, let us

denote the total branch length of the full phylogeny as B , and the total number of mutations happened on the focal site within the full phylogeny as k . On a branch i with length b_i , the number of mutations on the focal site follows a Poisson distribution with the expectation of kb_i/B . Thus, the probability of no change on branch i is $p_i = e^{-kb_i/B}$, and the probability of a change (which might be the result of more than one mutations) is $q_i = 1 - p_i = 1 - e^{-kb_i/B}$. We can further divide all branches within the phylogeny into two groups: those that have undergone genotype changes (denoted by G_1), and those that have not (denoted by G_0). Then the likelihood of observing the empirical data can be given by $L = \prod_{i \in G_1} q_i \prod_{j \in G_0} p_j = \prod_{i \in G_1} (1 - e^{-kb_i/B}) \prod_{j \in G_0} e^{-kb_j/B}$. Finally, GAMMA estimates k by finding a positive solution of equation $\partial \ln L / \partial k = 0$, since L is maximized (i.e. maximum likelihood) when the derivative of $\ln L$ is 0. As a result, each segregating site has its own ML-estimation of k , which is then divided by B and used as an approximation of per-site mutation rate (relative to other sites within the genome). Note that the unit of time for k is B , which is shared by all sites since they share the same full phylogeny. As a result, k values from different sites can be compared regardless of the actual value of B . Non-segregating sites were assumed to have a zero (negligible) mutation rate.

Second, the theoretical foundation of using polymorphism as an approximation of mutation rate is that the polymorphisms observed within the population of a species are predominantly neutral, or at least nearly-neutral. We have now discussed this point in detail in Method, which is pasted below. Note that in the third last sentence of this paragraph, we explained the calculation of MA-based mutation rate.

It is imperative to note that using DNA polymorphisms to estimate mutation rates is not ideal. There is a possibility that the observable level of DNA polymorphism, or mutation rate thereby estimated, may be affected by both mutation rate and natural selection, especially negative/purifying selection, as positive selection is generally much rarer. However, according to the neutral theory of molecular evolution⁴, most intraspecific polymorphisms are selectively neutral⁵, which is indeed empirically supported^{6,7}. For example, it was previously estimated that only 12% of coding SNPs found in yeast populations are deleterious⁷. Using Tajima's D test⁸, we also independently validated such general neutrality of polymorphisms in the current dataset of 190 strains by confirming that polymorphisms in the whole genome ($P = 0.1$ by Tajima's D test) and the vast majority of genes were indeed compatible with the neutral expectation (only 604 out of 6079 genes show a $P < 0.05$ by Tajima's D test, and none of them show a $P < 0.05$ when corrected for multiple testing using the Benjamini-Hochberg Procedure). Such general neutrality of polymorphisms dictates that the majority of variation in polymorphism levels can be attributed to variations in mutation rates rather than natural selection. Indeed, a higher level of polymorphism in a particular genomic region has often been used as an indicator of a higher local mutation rate⁹⁻¹¹. Moreover, we found that there is a positive correlation between our polymorphism-based mutation rates and those estimated from two previously published mutation accumulation datasets^{12,13} (Spearman's $\rho = 0.56$, $P < 0.05$. See Supplementary Fig. 11), which is generally accepted as the best direct estimate of mutation rate. Here the mutation rate of a gene was estimated by the total number of mutations

identified in the gene divided by the gene length¹⁴. This correlation further supports our view that polymorphism-based estimates are reasonable approximations for the mutation rate relative to other sites within the genome. Most importantly, even when we used mutation rate estimates from mutation accumulation lines, we are still able to detect the antimutator effect of nascent RNA structure (Supplementary Fig. 7).

We would like to highlight that this assumption is not only predicted by the neutral theory of molecular evolution^{4,5}, but have found general empirical support from previous studies^{6,7} and our used polymorphism dataset by Tajima's *D* test for neutrality. More specifically, using Tajima's *D* test, polymorphisms in the whole genome showed $P = 0.1$ (cannot reject neutrality). For individual genes, only 604 out of 6079 genes show a $P < 0.05$ by Tajima's *D* test, and none of them show a $P < 0.05$ when corrected for multiple testing using the Benjamini-Hochberg Procedure.

Third, our previous estimate was based on 190 yeast strains, not 109. In addition, we have repeated our analysis with polymorphisms found in 1011 yeast strains as downloaded from Peter et al.¹⁵, and with polymorphisms in four-fold degenerated sites. As a result, we continue to found the anticorrelation between nascent RNA structure and mutation rate. This figure below is the results from the 1011 yeast strains.

And this figure below is the results from the polymorphisms in four-fold degenerated sites.

Finally, and most importantly, the antimutator effect of nascent RNA structure is supported by the mutation rate derived from mutation accumulation lines (Supplementary Fig. 7), which is generally considered (by the reviewer too) as the best direct estimate of mutation rates (albeit with much rarer mutated sites compared to polymorphism data).

Comment 3

Figures 4d-f are also far from convincing. The authors give very low P values for the regressions, but they never say how the regressions were performed. For example, are they polynomial regressions, and if so, how many terms were included, and what was the statistical justification for adding terms? Why are the regressions truncated within the range of the data? One can see that for at least 50% of the range of data, the regressions are effectively flat, i.e., there is no response of the mutation rate to the dependent variable, and even if one takes into consideration the overall fits (which may be overdetermined), the model only explains 4% of the variation in the data.

Response:

The red lines in Figure 4d-f are LOESS (Locally Weighted Scatterplot Smoothing) lines, whereas the P values were from Spearman's Rank Correlation, and has nothing to do with the LOESS regression.

The LOESS lines were used to aid visual inspection. I.e. they serve only a representational purpose, and therefore could be eliminated without affecting our results or narrative. LOESS smoothed values near the extremities of x values were truncated since they display large fluctuations due to fewer data points available than in intermediate x values. More specifically, when a smoothing window size of 100 data points is used, a linear regression model will be fitted to the 100 nearest neighbors (in terms of x values) of a focal point (50 on either side). However, for data points with extreme x values, there will not be enough neighbors on either side, resulting in large errors in regression estimation. It is therefore necessary to truncate these regions of large errors. The P values in Figure 4d-f are statistical significance of the Spearman's Rank Correlation or the original data points, as indicated by the symbol of " ρ " in the figure, and in lines 238-243 of our manuscript, which is pasted below:

... structure indeed had lower R-loop scores (Fig. 4b and c). Notably, the corresponding correlations (Spearman's $\rho = -0.19$ and -0.35 , $P < 10^{-12}$ and 10^{-44} , respectively) were more than an order of magnitude stronger than previous observations made with computationally predicted nascent RNA structure¹⁴ (Spearman's $\rho = -0.014$). Then, after confirming the expected relationship between the spontaneous DNA mutation rate and the R-loop score (Fig. 4d), we also found that the prevalence of nascent RNA structure was anticorrelated with mutation rate (Spearman's $\rho = -0.22$ and -0.16 , $P < 10^{-19}$ and 10^{-10} , respectively, Fig. 4e and f), which were also much stronger than the prediction-based observation¹⁴ (Spearman's $\rho = -0.05$).

We agree that according to the correlation coefficients, the effect of nascent RNA folding did not appear to be strong. Nevertheless, correlations can be obscured by measurement errors and

confounded by other genomic factors, such as expression levels. The actual effect size, which is more biologically relevant, should be determined through manipulative experiments. These manipulation experiments were conducted using the CAN1 reporter, and indicated that strong nascent RNA folding could reduce mutation rates fivefold (from 2.5 to 0.5, relative mutation rates in arbitrary units). The related figure (Figure R7 in our previous response) is provided below.

Comment 4

On lines 250-252, the authors note regression coefficients of -0.18 and -0.09, and then conclude that 35% and 65% of the variance is explained by the model. I'm afraid that I have no idea as to how such a conclusion has been reached. The fraction of variance explained by a model is equal to the square of the correlation coefficient, 3.2 and 0.8% for the numbers given above. Perhaps I'm missing something here, but throughout the paper one can see by eye alone that the patterns being advocated are simply not very strong.

Response:

It is not that "35% and 65% of the variance is explained by the model", rather, it is that "approximately 35% and 66% of the antimutator effects of nascent RNA folding were mediated by the R-loop". The logic was as follows: when the R-loop signal is not controlled, the correlation between nascent folding and the mutation rate is -0.219 (fraction of variance explained = 4.80%), and when the R-loop signal is controlled, the correlation is -0.177 (fraction of variance explained = 3.13%). According to the difference between the two correlations, controlling R-loop reduces the fraction of variance explained from 4.84% to 3.13%, a reduction of 34.8% (= 1-3.13%/4.80%). Therefore, this "35%" is the fraction of antimutator effects of nascent RNA that is mediated by R-loop. We have clarified this part as below in our manuscript as below.

To further investigate the role of the R-loop in mediating the antimutator effects of the nascent RNA structure, we calculated the partial correlation between the prevalence of nascent RNA structure and mutation rate, controlling the R-loop score. These partial

correlations were $\rho = -0.177$ ($P < 10^{-11}$) for the negative single-stranded score and $\rho = -0.086$ ($P < 10^{-3}$) for the Gini index of the single-stranded score, which respectively suggested that approximately 34.8% ($= 1 - (-0.177^2/-0.219^2)$) and 69.6% ($= 1 - (-0.086^2/-0.156^2)$) of the antimutator effects of the nascent RNA folding were mediated via the R-loop (Fig. 4g).

Comment 5

It is also difficult to know how much faith to put into the Can1 reporter-construct results referred to from the supplemental table 8. The problem here is that reporter-construct mutation-rate estimates are known to be only weakly correlated with overall genomic mutation rates, and here we have the potential additional problem that the constructs produced by the authors may have effects on the mutation rate completely independent of rna structural effects.

Response:

Our hypothesis is that the nascent RNA folding would affect the mutation rate of nearby genomic regions (See Figure 4a). In other words, the nascent RNA folding is a LOCAL regulator of mutation rate, not a GLOBAL regulator. The global mutation rate was irrelevant here. It should be stressed that our assay with CAN1 was a manipulative experiment using a standard case-control design in which we altered the folding of the nascent RNA and observed the expected changes in mutation rates as a result. We further showed that this effect was mediated via the R-loop based on the overexpression of RNaseH1.

Comment 6

On lines 305-308, the authors suggest that they compared the evolutionary conservation of sequences to structural features. However, although they suggest that the approach used to estimate the degree of conservation is in the Methods section, I could not find it. They may be referring to the polymorphism data, but it is well known that polymorphisms are enriched for variants that never go to fixation, so that such indices are not appropriate measures of functional constraint. A more appropriate approach is to use between-species divergence data, ideally normalize by polymorphism data (e.g., as a neutrality index).

Response:

There is a section in our Methods titled “Expression levels and evolutionary conservation of *S. cerevisiae* genes”, which is pasted below

The expression level of the yeast transcriptome was extracted from a previous RNA-seq-based report¹⁶. Evolutionary conservation was estimated inversely by the ratio between the number of nonsynonymous substitutions per nonsynonymous site (dN) and the number of synonymous substitutions per synonymous site (dS) detected from one-to-one orthologs between *S. cerevisiae* and *Saccharomyces bayanus* following previously described pipelines¹⁷.

We used dN/dS derived from inter-specific comparisons as the measure of evolutionary conservation, not polymorphism.

Comment 7

Finally, the cancer data outlined in Figures 6a-f are even more unconvincing in terms of identifying a mutational effect. Again, the authors fit a model with apparently at least three (based on the inflections in the curvilinearity (and perhaps more) coefficients, with the overall fits being extremely flat.

Response:

This is similar to comment 4 above. All of these results are based on Spearman's Rank Correlation, not regression.

Reference

1. Gu, X. & Zhang, J. A simple method for estimating the parameter of substitution rate variation among sites. *Molecular biology and evolution* **14**, 1106-1113 (1997).
2. Maclean, C.J., *et al.* Deciphering the Genic Basis of Yeast Fitness Variation by Simultaneous Forward and Reverse Genetics. *Molecular biology and evolution* **34**, 2486-2502 (2017).
3. Zhang, J. & Nei, M. Accuracies of ancestral amino acid sequences inferred by the parsimony, likelihood, and distance methods. *J Mol Evol* **44 Suppl 1**, S139-146 (1997).
4. Kimura, M. *The neutral theory of molecular evolution*, (Cambridge University Press, 1983).
5. Kimura, M. The neutral theory of molecular evolution and the world view of the neutralists. *Genome* **31**, 24-31 (1989).
6. Bowcock, A.M., *et al.* Drift, admixture, and selection in human evolution: a study with DNA polymorphisms. *Proceedings of the National Academy of Sciences of the United States of America* **88**, 839-843 (1991).
7. Doniger, S.W., *et al.* A catalog of neutral and deleterious polymorphism in yeast. *PLoS Genet* **4**, e1000183 (2008).
8. Tajima, F. Statistical method for testing the neutral mutation hypothesis by DNA polymorphism. *Genetics* **123**, 585-595 (1989).

9. Lercher, M.J. & Hurst, L.D. Human SNP variability and mutation rate are higher in regions of high recombination. *Trends Genet* **18**, 337-340 (2002).
10. Aggarwala, V. & Voight, B.F. An expanded sequence context model broadly explains variability in polymorphism levels across the human genome. *Nat Genet* **48**, 349-355 (2016).
11. Ellegren, H., Smith, N.G. & Webster, M.T. Mutation rate variation in the mammalian genome. *Curr Opin Genet Dev* **13**, 562-568 (2003).
12. Fares, M.A., Keane, O.M., Toft, C., Carretero-Paulet, L. & Jones, G.W. The roles of whole-genome and small-scale duplications in the functional specialization of *Saccharomyces cerevisiae* genes. *PLoS Genet* **9**, e1003176 (2013).
13. Zhu, Y.O., Siegal, M.L., Hall, D.W. & Petrov, D.A. Precise estimates of mutation rate and spectrum in yeast. *Proceedings of the National Academy of Sciences of the United States of America* **111**, E2310-2318 (2014).
14. Chen, X., Yang, J.-R. & Zhang, J. Nascent RNA folding mitigates transcription-associated mutagenesis. *Genome Research* **26**, 50-59 (2016).
15. Peter, J., *et al.* Genome evolution across 1,011 *Saccharomyces cerevisiae* isolates. *Nature* **556**, 339-344 (2018).
16. Nagalakshmi, U., *et al.* The transcriptional landscape of the yeast genome defined by RNA sequencing. *Science* **320**, 1344-1349 (2008).
17. Zhang, J. & Yang, J.R. Determinants of the rate of protein sequence evolution. *Nature reviews. Genetics* **16**, 409-420 (2015).

REVIEWER COMMENTS

Reviewer #1 (Remarks to the Author):

The authors have done a lot of revisions for this manuscript. However, I still have some remaining questions:

- 1) RT stop densities are problematic as a way to measure RNA structure information because there can be random stoppage sites, as such the replicates can appear to be artificially highly correlated (such as in Fig 1b). As such, we and others usually subtract NAI RT stop densities with DMSO RT stop densities to generate SHAPE reactivities. How do the results look after subtracting the DMSO reads?
- 2) For Figure 3a, and 4h, why is the in vivo NAI-RT stoppages flat for most of the genes?
- 3) I still think that the current coverage for measuring RNA structure in this manuscript is low- the authors should report the data with a higher threshold.
- 4) For nascent RNA sequencing- how many non-full length reads are they able to obtain for the majority of the mRNAs in the cell?
- 5) There are around 200 transcripts with intron in *S. cerevisiae*. Could the presence of the intron (preprocessing) affect RNA structure and cause the difference between nascent and mature transcripts?

Reviewer #4 (Remarks to the Author):

This study introduces two key advancements. Firstly, the researchers successfully adapted the prokaryotic SPET-seq technique for use in eukaryotes, making necessary modifications and protocol improvements. Within this section, the authors applied the newly developed eSPET-seq method to both wildtype and genetically modified yeast strains, demonstrating good repeatability across replicates. Additionally, some of the sequencing results were validated through DRIP-RT-qPCR. Based on the data quality of eSPET-seq and the reliability of the nascent RNA structure, the findings appear promising.

Secondly, and perhaps more intriguingly, the study reveals a correlation and potential causality between nascent RNA structure and the spontaneous mutation rate of DNA. Given that mutation rate is a critical parameter in the fields of evolution and genetics, any genomic features or molecular traits that directly impact or associate with mutation rate hold significant value and warrant reporting.

To examine the authors' hypothesis, they employed multiple datasets to estimate the mutation rate:

1. Polymorphism-based estimation, which, while not representing the actual mutation rate, has been shown in previous studies to exhibit strong correlations with the density of polymorphisms and the true mutation rate across the genome.
2. Mutations identified through mutation-accumulation experiments, which serve as the gold standard for estimating mutation rate.
3. Mutations observed in tumors, representing real mutations that arise during somatic development.
4. The CAN1-based fluctuation test, a well-established method for estimating the mutation rate of a single locus. This test, developed by Nobel laureates Joshua Lederberg and Salvador Luria, demonstrates that the occurrence of mutations is independent of their environment.

All of the aforementioned datasets consistently support the notion that nascent RNA structure influences mutation rate.

In my view, the authors of this paper have presented an interesting hypothesis supported by multiple

independent datasets. Moreover, conducting such a comprehensive study undoubtedly required a tremendous amount of effort. Consequently, I highly recommend accepting this manuscript with minor modifications.

Detailed comments:

1. I suggest integrating Supplementary Fig. 8 into the main text. The results based on polymorphism, mutation accumulation, and tumor mutations represent associations rather than causality. Conversely, manipulative experiments involving *CAN1* provide evidence suggesting a potential causal relationship between nascent RNA structure and mutation rate, making it a crucial aspect of this study.
2. Another study (PMID: 31056389) has conducted mutation accumulation experiments on a larger scale than the reference 33. I recommend incorporating this dataset into your analysis.
3. It is more common to use "dN dS" (italic d and subscript N and S) instead of "dN dS" in scientific writing.

Point-by-Point Response to Reviewer's Comments

Reviewer #1 (Remarks to the Author):

Overall Comment

The authors have done a lot of revisions for this manuscript. However, I still have some remaining questions:

Response:

Our sincere gratitude goes out to the reviewer for taking a lot of time to review our manuscript.

Comment 1

RT stop densities are problematic as a way to measure RNA structure information because there can be random stoppage sites, as such the replicates can appear to be artificially highly correlated (such as in Fig 1b). As such, we and others usually subtract NAI RT stop densities with DMSO RT stop densities to generate SHAPE reactivities. How do the results look after subtracting the DMSO reads?

Response:

We would like to thank the reviewer for his question. We understand that the reviewer is concerned about reverse transcription (RT) stops not due to NAI-N₃ modification. Most of our results, including figures 3/4/5/6, evaluate SHAPE reactivity or single-strandedness by contrasting RT stop densities in NAI-N₃-treated samples with those in paired DMSO-treated samples. This has been explained in Methods by the formula for the single-stranded score.

$$\theta_{NAI-N_3}(i) = \log_2\left(\frac{N_{NAI-N_3}(i) + 1}{N_{DMSO}(i) + 1} + 1\right)$$

According to the formula, single-stranded score contrasts NAI-N₃ samples with paired

DMSO samples, which is consistent with the reviewer's suggestion. In the following paragraphs, we discuss the justification for directly using NAI-N₃ RT stop densities in the two remaining figures 1 and 2.

In principle, NAI-independent RT stoppage can broadly be categorized into two types: (i) sequence context-dependent, such as those induced by extreme GC%, and (ii) sequence context independent, which is due to molecular stochasticity. It is important to note that, among these two types of NAI-independent stoppage, the context-independent stop cannot create biological signals, but rather can only obscure them. In contrast, the context-dependent stop could potentially bias comparisons of different genes or different sites, and for this reason, should be properly accounted for, as suggested by the reviewer.

As stated above, RT stop (or its densities) is only used on two occasions in our manuscript. FIRST, raw sequencing data is compared between biological replicates, as shown in Figures 1b and noted by the reviewer. The correlation for Figure 1b remained high even when we used single-stranded scores instead of RT stops, as illustrated in Figure RRR1 below.

Figure RRR1

Furthermore, Figure 1 is intended to offer the reader an overview before any data processing is conducted and to demonstrate that the experiment is repeatable. No biological conclusion was drawn from Figure 1. Therefore, we consider it appropriate to use raw RT stop counts in this instance.

SECOND, RT stop densities are used to measure co-transcriptional folding, as shown in figure 2. This analysis compares different transcriptional intermediates with

different lengths for the RT stop density of the same nucleotide, which should have the exact same sequence context. This comparison should, therefore, have canceled out the effect of context-dependent RT stoppage, thus avoiding the creation of false biological signals.

Comment 2

2) For Figure 3a, and 4h, why is the in vivo NAI-RT stoppages flat for most of the genes?

Response:

We thank the reviewer for the question. The reviewer's comment seems to pertain to Figure 3c rather than Figure 3a, according to the description. As stated above, Figures 3c and 4h show the single-stranded score. This is not the raw NAI RT stoppage, but the contrast between the NAI RT stop densities and the DMSO RT stop densities in paired samples. It can be seen in Figures 3c and 4h that the extreme values within other regions of the gene inflate the value range covered by the y axis, compressing the variations within the "flat" regions. Figure RRR2a (YPL250C, corresponding to Figure 3c) and b (YDR110W, corresponding to Figure 4h) below demonstrate this by zooming in on a particular region of the y axis. In fact, Figure RRR2 shows that the flat region exhibits clear ups and downs, indicating that the experimental results are informative. Moreover, these "flat" regions exhibit low single-stranded scores in comparison to other regions within the gene, suggesting that they have weaker SHAPE reactivity compared to other sub-genic regions within the gene, but not necessarily compared to other genes.

Figure RRR2

Comment 3

3) I still think that the current coverage for measuring RNA structure in this manuscript is low- the authors should report the data with a higher threshold.

Response:

We appreciate the reviewer's comments. After much consideration, we have decided to stay with the current reporting threshold of one RT stop per nucleotide (1 stop/nt) . The following are our reasons for making this decision.

The canonical definition of read coverage is the total number of nucleotides in all reads divided by the size of the targeted region, or in other words, the average number of reads covering a nucleotide. The threshold of 1 stop/nt is, however, based on the number of reads and does not take into account the number of nucleotides within each read. This is illustrated in the scatter plot Figure RRR3 below by displaying a dot for each gene, with the average number of RT stops per nucleotide on the x axis and the average read coverage on the y axis.

Figure RRR3. Scatter plot of average read coverage (y) versus average RT stop per nucleotide (x). Each dot represents one gene.

It is evident from the figure that our threshold of 1 stop/nt actually corresponds to a read coverage of 30. These genes above the 1 stop/nt threshold have a median read coverage of 110. Based on these metrics, it can be concluded that the threshold chosen is sufficiently high. Given the nature of eSPET-seq, where only the 4' and 3' ends of the reads are informative, it makes more sense to define a threshold with the number of reads rather than nucleotides. We have added this Figure RRR3 to Supplemental Figure 1 in the revised manuscript.

Additionally, a previous study [Ding, Y. et al. In vivo genome-wide profiling of RNA secondary structure reveals novel regulatory features. Nature 505, 696-700 (2014)] has suggested that a threshold of 1 stop/nt is adequate for reliable structure determination. Furthermore, a threshold of 1 stop/nt seems more natural than other numbers such as 5 or 10.

More importantly, we found that reporting data with a higher coverage threshold decrease number of samples (genes) and thereby statistical power, but did not alter any of our conclusions. Our previous responses demonstrated this by showing results with higher thresholds. These previous responses are quoted below.

Per the reviewer's suggestion, we've tried more stringent cutoffs. All results hold, as shown below. Unless otherwise stated, the results are listed for each relevant result figure panel, with the **left-most** result being the one from the main figure using the threshold of **1 stop/nt**, and the **middle** and **right-most** results from the thresholds of **2 stop/nt** and **4 top/nt**, respectively.

Fig.3a. Structural similarity between nascent and mature RNA. Most genes displayed significant similarity, although the number of genes reduced due to increased threshold.

Fig.3d. The structural similarity between mature RNAs folded *in vivo* and *in vitro* was compared to that between nascent and mature RNAs folded *in vivo*. Regardless of the threshold used, the results showed a consistent positive correlation.

Fig.3e. Standard boxplot showing that the top 50% of genes with higher structural

similarity between mature RNAs folded *in vivo* and *in vitro* tended to have less RBP binding sites than the other genes. Regardless of the threshold used, the results remained significant by Mann-Whitney test.

Fig.3f. Standard boxplots showing that the top 50% of genes with higher structural similarity between nascent and mature RNAs folded in vivo tended to have less RBP binding sites than the other genes. Regardless of the threshold used, the results remained significant by Mann-Whitney test.

Fig.4b. The negative of single-stranded score was compared to the R-loop score of the same gene. Regardless of the threshold used, the results showed a consistent negative correlation.

Fig.4c. The Gini index of single-stranded score was compared to the R-loop score of the same gene. Regardless of the threshold used, the results showed a consistent negative correlation.

Fig.4e. The negative of single-stranded score was compared to the mutation rate of the same gene. Regardless of the threshold used, the results showed a consistent negative correlation.

Fig.4f. The Gini index of single-stranded score was compared to the mutation rate of the same gene. Regardless of the threshold used, the results showed a consistent negative correlation.

Fig.4g. The R-loop-dependent fraction of antimutator effect by nascent RNA folding was estimated by contrasting the correlation and the partial correlation (with R-loop score controlled) between the mutation rate and the prevalence of nascent RNA structure. The fraction of R-loop dependent antimutagenic effects by negative of and Gini index of single-stranded score ranged from 35% to 54% and 68% to 98% ,respectively, across the three threshold conditions. These results consistently supported our conclusion that a large fraction of the antimutator effect of nascent folding was mediated via the R-loop.

Fig.4i. Three odds ratios representing the correspondence among single-stranded scores, R-loop scores and mutation rates (OR_{S-R} , OR_{R-M} and OR_{S-M}) were calculated for each gene, and then combined and tested for statistical significance by the Cochran-Mantel-Haenszel chi-squared test. Regardless of the threshold used, the results significantly supported our hypothesized model of nascent RNA folding mitigating TAM by dissolving the R-loop.

Fig.4j. Three within-gene correlations among negative of single-stranded score, R-loop score and mutation rate of nucleotide positions were averaged (red arrow) and compared with their random expectations (histogram), which were estimated by permutating the negative of single-stranded score, R-loop score and mutation rate within each gene 1,000 times. P values from permutation tests are indicated. The three panels at the top represent results from 1 stop/nt, while those at the middle represent results from 2 stops/nt, and those at the bottom represent results from 4 stops/nt. Regardless of the threshold used, the results significantly supported our hypothesized model of nascent RNA folding mitigating TAM by dissolving the R-loop.

Fig. 5a. The expression level of a gene was compared to the negative of single-stranded score of the same gene. Regardless of the threshold used, the results showed a consistent positive correlation.

Fig. 5b. The expression level of a gene was compared to the Gini index of single-stranded score of the same gene. Regardless of the threshold used, the results showed a consistent positive correlation.

Fig. 5c. The evolutionary conservation of a gene was compared to the negative of single-stranded score of the same gene. Regardless of the threshold used, the results showed a consistent positive correlation.

Fig. 5d. The evolutionary conservation of a gene was compared to the Gini index of single-stranded score of the same gene. Regardless of the threshold used, the results showed a consistent positive correlation.

In summary, all the results and conclusions are robust to more stringent cutoffs.

Finally, although we consider 1 stop/nt to be a natural and appropriate threshold, there may be readers, similar to the reviewer, who wish to know the results at higher thresholds. Therefore, we updated the Github repository associated with our study (<https://github.com/GongwangYu/NascentRNAfolding>) to include the sequencing coverage in the processed data, and explicitly announced a “coverage_threshold” variable, which will be applied globally across the entire analysis script. By changing the value of "coverage_threshold" and running the script again, readers may easily obtain the results with a different threshold.

Comment 4

4) For nascent RNA sequencing- how many non-full length reads are they able to obtain for the majority of the mRNAs in the cell?

Response:

We appreciate the reviewer's question. As our eSPET-seq method is capable of capturing the 3'-end of each nascent transcript, we can identify whether a particular pair of reads originates from a non-full length (nascent) or full length (mature) mRNA. In particular, any pair of reads whose Read 2 (representing the transcription site. See Figure 1a) is downstream of the polyA signal of the mRNA can be considered derived from full-length mRNA, while the rest can be considered derived from non-full-length

mRNA. In Figure RRR4, we present a histogram of the number of reads derived from non-full-length mRNA for each gene (above the 1 stop/nt threshold) in the NAI-N3-treated samples. For these genes, the average number of non-full-length reads per gene is 3008.

Figure RRR4. Histogram for number of non-full-length reads from each gene.

Comment 5

5) There are around 200 transcripts with intron in *S. cerevisiae*. Could the presence of the intron (preprocessing) affect RNA structure and cause the different between nascent and mature transcripts?

Response:

We thank the reviewer for the helpful suggestion. We examined the influence of introns on RNA structure by extracting intron-containing genes with 1 average RT stop per nucleotide from our eSPET-seq and icSHAPE datasets. A total of 110 genes were selected as a result. Each gene's single-stranded scores were further normalized to an average value of 1 individually for nascent and mature RNA. Following that, the single-stranded scores of 100 nt upstream and downstream of the intron are collected for both nascent and mature RNA, and compared as shown in Figure RRR5A. To summarize similar comparisons for the 110 genes, we provided a volcano plot in Figure RRR5B to illustrate the structural difference between nascent and mature transcripts, in which each dot represents an intron-containing gene, while x axis represents the ratio of

average single-stranded score (nascent divided by mature) and y axis represents statistical significance by Wilcoxon signed-rank test.

Figure RRR5. The impact of intron on RNA secondary structure. **(A)** An intron-containing gene (YNL112W) was used to demonstrate our method for comparing nascent and mature RNA structure. The single-stranded scores of each gene were first normalized to an average value of 1 individually for nascent and mature RNA. The normalized single-stranded scores of 100 nt upstream and downstream of the intron in both nascent and mature RNA (dashed boxes) are collected and compared. **(B)** Volcano plot showing the difference in average single-stranded scores between nascent and mature RNAs. A dot represents an intron-containing gene, while the x axis represents the ratio of average single-stranded score (nascent divided by mature) and the y axis represents statistical significance according to the Wilcoxon signed-rank test. The genes whose nascent RNAs exhibit higher or lower average single-stranded scores than mature RNAs ($P < 0.05$, Wilcoxon signed rank test) are colored red or green, respectively. Colored numbers indicate the number of genes within each category. Genes with insignificant differences are indicated by gray dots.

In spite of these findings, we have decided not to include them in our manuscript. As the reviewer pointed out, yeast contains a relatively small number of intron-

containing genes. A more comprehensive investigation of the impact of introns on nascent RNA folding or its difference with mature RNA would require the use of species with more intron-containing genes, such as mice. Indeed, as shown in Figure RRR5, there is not enough sample size to find statistical significance regarding whether introns result in increased or decreased RNA folding (Binomial $P > 0.05$ for 41 versus 27 in Figure RRR5). We are not suggesting that there would be a significant result. It is believed, however, that more than 200 intron-containing genes are needed to reach a more comprehensive conclusion, and such a study should be conducted separately.

Reviewer #4 (Remarks to the Author):

Overall Comments

This study introduces two key advancements. Firstly, the researchers successfully adapted the procaryotic SPET-seq technique for use in eukaryotes, making necessary modifications and protocol improvements. Within this section, the authors applied the newly developed eSPET-seq method to both wildtype and genetically modified yeast strains, demonstrating good repeatability across replicates. Additionally, some of the sequencing results were validated through DRIP-RT-qPCR. Based on the data quality of eSPET-seq and the reliability of the nascent RNA structure, the findings appear promising.

Secondly, and perhaps more intriguingly, the study reveals a correlation and potential causality between nascent RNA structure and the spontaneous mutation rate of DNA. Given that mutation rate is a critical parameter in the fields of evolution and genetics, any genomic features or molecular traits that directly impact or associate with mutation rate hold significant value and warrant reporting.

To examine the authors' hypothesis, they employed multiple datasets to estimate the mutation rate:

1. Polymorphism-based estimation, which, while not representing the actual mutation rate, has been shown in previous studies to exhibit strong correlations with the density of polymorphisms and the true mutation rate across the genome.
2. Mutations identified through mutation-accumulation experiments, which serve as the gold standard for estimating mutation rate.
3. Mutations observed in tumors, representing real mutations that arise during somatic development.
4. The CAN1-based fluctuation test, a well-established method for estimating the mutation rate of a single locus. This test, developed by Nobel laureates Joshua Lederberg and Salvador Luria, demonstrates that the occurrence of mutations is independent of their environment.

All of the aforementioned datasets consistently support the notion that nascent RNA structure influences mutation rate.

In my view, the authors of this paper have presented an interesting hypothesis supported by multiple independent datasets. Moreover, conducting such a comprehensive study undoubtedly required a tremendous amount of effort. Consequently, I highly recommend accepting this manuscript with minor modifications.

Response:

We are deeply grateful for the reviewer's constructive feedback and kind words. We appreciate his/her recognition of our efforts and the time taken to review our work.

Comment 1

I suggest integrating Supplementary Fig. 8 into the main text. The results based on polymorphism, mutation accumulation, and tumor mutations represent associations rather than causality. Conversely, manipulative experiments involving CAN1 provide

evidence suggesting a potential causal relationship between nascent RNA structure and mutation rate, making it a crucial aspect of this study.

Response:

We appreciate the thoughtful comments made by the reviewer. Supplementary Figure 8 has been added to the revised manuscript as Figure 5.

Comment 2

Another study (PMID: 31056389) has conducted mutation accumulation experiments on a larger scale than the reference 33. I recommend incorporating this dataset into your analysis.

Response:

We would like to thank the reviewer for this very constructive suggestion. In response to the reviewer's suggestion, we collected the mutations from this reference, and compared the prevalence of nascent RNA structures in regions with and without mutations. We found that non-mutated sites exhibit a significantly higher prevalence of RNA structures than mutated sites in the same gene (Figure RRR6 below). It is consistent with results obtained from two other mutation accumulation lines previously used in our manuscript, which further supports the antimutator effect of nascent RNA folding that we proposed in our manuscript. This result has been included in Supplemental Figure 7 in the revised manuscript.

Figure RRR6. Violin plot for prevalence of nascent RNA secondary structure in

mutated versus non-mutated sites in mutational accumulation experiments from a study (PMID: 31056389). The prevalence of nascent RNA secondary structure was approximated by either negative value (left panel) or Gini index (right panel) of single-stranded scores. The black dot inside the violin shows the mean. *P*-values are based on Wilcoxon signed-rank tests.

Comment 3

It is more common to use "d_N d_S" (italic d and subscript N and S) instead of "dN dS" in scientific writing.

Response:

Corrected as suggested. Thank you !

REVIEWERS' COMMENTS

Reviewer #1 (Remarks to the Author):

The authors have addressed my concerns. I have no other questions for now.

Reviewer #4 (Remarks to the Author):

The results based on the newly added data have strengthened the original conclusions. Therefore, I am satisfied with this revision and do not have any further questions.

Point-by-Point Response to Reviewer's Comments

Reviewer #1 (Remarks to the Author):

Overall Comment

The authors have addressed my concerns. I have no other questions for now.

Response:

We are grateful to the reviewer for his or her time and support in reviewing our manuscript.

Reviewer #4 (Remarks to the Author):

Overall Comments

The results based on the newly added data have strengthened the original conclusions. Therefore, I am satisfied with this revision and do not have any further questions.

Response:

The reviewer's insightful comments, constructive suggestions, and kind support are deeply appreciated.